# Optimization of Shelter Location Based on a Combined Static/Dynamic Two-Stage Optimization Methodology: A Case Study in the Central Urban Area of Xinyi City, China

**Guangchun Zhong [1], Guofang Zhai [1] and Wei Chen [2,\*]**

1   School of Architecture and Planning, Nanjing University, Nanjing 210093, China;
    gczhong@smail.nju.edu.cn (G.Z.); guofang_zhai@nju.edu.cn (G.Z.)
2   School of Geographic and Biologic Information, Nanjing University of Posts and Telecommunications,
    Nanjing 210023, China
\*   Correspondence: chen_wei@njupt.edu.cn; Tel.: +86-19921316479

**Abstract:** Determining how to reasonably allocate shelters in the central area of the city and improve evacuation efficiency are important issues in the field of urban disaster prevention. This paper introduces the methodology and mathematical model from the field of crowd emergency evacuation to shelter location optimization. Moreover, a shelter location optimization method based on the combination of static network analysis and dynamic evacuation simulation is proposed. The construction costs and evacuation times are taken as the objective functions. In the first stage, based on the static network analysis, a circular evacuation allocation rule based on the gravity model is proposed, and the genetic algorithm is then designed to solve the feasible schemes with the lowest shelter construction costs. In the second stage, the evacuation time is taken as the optimization objective. The age differences of refugees, the selection of evacuation routes, and the behavior of adults helping children and the elderly are simulated in a dynamic evacuation simulation model. The traditional social force model is improved to conduct a regional evacuation simulation and determine the optimal scheme with the shortest evacuation time. Finally, the central urban area of Xinyi City, Jiangsu Province, China, is taken as an empirical case.

**Keywords:** shelter; static/dynamic analysis; social force model; genetic algorithm; two-stage multi-objective optimization

## 1. Introduction

Shelters are typical spaces and buildings equipped with emergency support infrastructure, emergency auxiliary facilities, and emergency support equipment and materials for the living security of refugees and centralized rescue after disasters [1]. With the development of natural and social systems becoming increasingly more complex, the frequency and types of disasters are increasing. Risk has three main components: the occurrence of a disastrous event, vulnerability, and exposure. This study focuses on actions to reduce the exposure component. The main macro-action to reduce exposure is evacuation, which consists of reducing the number of users and goods that can experience negative effects when emergency events occur [2]. Thus, determining how to scientifically allocate and optimize the location of shelters has become of increasing concern. A scientific shelter location scheme can significantly reduce property losses and casualties, and can improve the efficiency and safety of the evacuation process after disasters.

Scholars have carried out a series of investigations on the location of shelters, and have put forward the P-center model [3], the P-median model [4], the set covering model [5], and the maximum covering model [6]. Moreover, based on the classical model, distance constraints [7] and capacity constraints [8] have been added. The types of objective functions can be divided into single-objective and multi-objective models. To simplify the problem,

some scholars have adopted a single-objective model to establish the location model in the early research stage [9,10], and then adopted the minimization of the construction cost or the number of shelters as the optimization objective. However, due to the complexity of the location optimization of shelters, scholars became unsatisfied with the establishment of single-objective optimization models. Thus, multi-objective optimization models have been established by integrating multiple optimization objectives, such as the construction cost [11], the evacuation distance [12], time [13], the safety of evacuation roads [14], and disaster risks [15–17]. Different from single-objective location models, the solution of multi-objective location models is the balance of multiple objectives, and the number of solutions is usually more than one. There exist three general methods by which to solve the multi-objective optimization model of shelters: (1) the conversion of the multi-objective optimization model into a single-objective optimization model via weighting [18]; (2) the transformation of the multi-objective shelter optimization problem into a two-stage single-objective optimization problem [19,20]; (3) the adoption of the heuristic algorithm to solve the Pareto optimal solution [16,21]. According to the different levels, internal characteristics, classifications, and corresponding functions of different shelters, scholars have put forward multi-level shelter location models [21–23]. For instance, based on the existing shelters in the community and evacuation demands in different periods, Ye [24] proposed an evacuation planning method based on multiple scenarios at the community level. Some scholars have also incorporated the distribution of relief materials [25] [26] and disaster process simulation [8,27] into shelter location modeling. The optimization models of shelter location have experienced changes from simple to complex, from a single factor to a variety of complex urban environmental factors, from the macro-level to the micro-level, and from the single level to multiple levels. These developments have made shelter location models more detailed, which provides a feasible theoretical basis for practical work.

The optimization algorithms for solving the location optimization models of shelters are divided into two categories, namely, accurate and approximate algorithms. When the shelter location model is a single-objective optimization problem based on a polynomial function or a multi-objective optimization problem that can be transformed into a single-objective function via the objective weighting method, objective programming method, and efficacy coefficient method, precise optimal solutions can be determined by accurate algorithms, such as the branch-bound algorithm [28], the simplex method, and the Lagrange relaxation method [29]. The solution process is implemented via optimization software such as Lingo and CPLEX. However, the location of shelters with multi-objective constraints is an NP-hard problem, which is difficult to be solved by accurate algorithms. Therefore, some scholars have adopted intelligent optimization algorithms, such as the genetic algorithm [30], simulated annealing algorithm [31], particle swarm algorithm [12,32], ant colony algorithm [33], and tabu search algorithm [34], to solve shelter location models with multi-objective constraints. Finally, the visualization of the optimal scheme is implemented on the GIS platform [35].

The studies described previously are based on the static network analysis method, and the proposed models are, therefore, characterized by the following deficiencies in terms of the evacuation demand calculation and evacuation allocation rules:

(1) Failure to satisfy the evacuation demands in peak hours. The evacuation demands in previous research are determined based on a permanent population [27,36]. However, due to the separation of workplaces and residences in central urban areas, there are substantial differences in the population distributions during daytime and nighttime [37–39]. The evacuation demands determined by a permanent population cannot reflect the dynamic spatiotemporal distribution of the population, and cannot satisfy the evacuation demands during the peak period; and

(2) Failure to consider the crowd evacuation process and evacuation behavior. The current optimization models usually assume that pedestrians at one demand point will follow the arrangement of the government and go to their designated shelter [40], which is inconsistent with the authentic evacuation process. The evacuation behavior and psychol-

ogy of pedestrians, as well as the time-varying changes of road congestion, have not been considered previously [41].

In terms of the simulation of crowd emergency evacuation, evacuation models can be divided into micro-, macro-, and meso-models [42]. From the perspective of the evacuation allocation method, evacuation models can be divided into dynamic and static models. Dynamic models indicate that pedestrians or vehicles select their movement direction at each intersection, while static models indicate that pedestrians or vehicles follow the pre-designed route without changing their evacuation routes midway [43]. Macro-models [44] regard the movement of people as fluid, and use the partial differential equations in fluid dynamics to describe the time-varying changes of the evacuation velocity and density [45]. These models have high computational efficiency and do not consider the details of individual behavior, but transform the constraints from the road network into network flow for optimization. However, macro-models update the state variables at regular intervals via the dynamic state equation, which cannot reflect the interaction and heterogeneity between pedestrians. Meso-models are between micro- and macro-models. The discrete simulation method is adopted to track the movement of pedestrians in the evacuation process. Meso-models are not only characterized by a reduced amount of calculation, but also record a portion of the important details in the evacuation process [46]. Finally, micro-models regard people as individual particles. These models can reflect differences between pedestrians, and the movement description is more accurate and natural. Common micro-models include the social force model [47], the cellular automata model [48,49], and the multi-agent model [50,51], among others. The simulation of user behaviour during an evacuation is a very complex problem, as the behaviour depends on different factors, such as kind and entity of the dangerous event, the socio-economic characteristics of users, and panic [52]. The social force model has been widely used because of its high accuracy and strong continuity. The specific evacuation behavior of pedestrians, such as herd behavior [50], following behavior [53], the fast or slow effect, and the leader effect [54], can be simulated by the social force model. Moreover, the social force model can be combined with the multi-agent model [55,56] to simulate the heterogeneity of different categories of pedestrians. Therefore, the social force model is adopted in the present study to conduct a regional evacuation simulation and, ultimately, scientifically and accurately optimize the location of shelters.

At present, the research on dynamic evacuation simulation is concentrated on large public buildings [57], teaching buildings [58], subways [59], and high-rise buildings [60], and evacuation simulation on the community scale is rare. Previous regional evacuation simulations did not consider the spatial heterogeneity of the age of pedestrians [61], which has led to a large deviation of the simulation results from authentic evacuation scenarios. Former studies have shown that the evacuation velocity of pedestrians is significantly correlated with age [62]. The age structure varies greatly with the types of land use and industries in the space, and there are significant differences in the age structures of different plots. For example, the proportion of elderly people in commercial districts is small, while that in old communities is high. The overall evacuation efficiency of plots with a high proportion of elderly people is low, which will influence the overall evacuation process and the spatial configuration of shelters.

Therefore, this study introduces the methodology and model of emergency evacuation simulation into the location optimization of shelters. Firstly, a regional emergency evacuation model is established, which considers the choice of evacuation routes by pedestrians. Three types of pedestrian agents, including the elderly, adults, and children, are established, and the behavior of adults helping the elderly and children during an evacuation is simulated. Second, by combining the advantages of the traditional location optimization model based on static network analysis and dynamic emergency evacuation simulation, this study proposes a combined static/dynamic two-stage location optimization methodology for which the number of shelters and the total evacuation time are taken as the optimization objectives. The objective function of the optimization in the first stage is the

construction cost of shelters, and circular evacuation allocation rules are proposed based on the gravity model. The genetic algorithm is then designed to solve the model. The optimization in the second stage establishes the regional dynamic evacuation model of the feasible schemes of the first stage. The total evacuation time of each feasible scheme is then compared to determine the optimal location scheme with the shortest evacuation time. Finally, Xinyi City is taken as an empirical case to verify the feasibility and applicability of the proposed optimization methodology based on the combination of static analysis and dynamic simulation.

The proposed methodology of this study makes four contributions: (1) introduces the methodology and mathematical model from the field of crowd emergency evacuation to shelter location optimization. The deficiencies in the configuration of urban shelters can be intuitively found; (2) the combined static/dynamic optimization methodology cannot only efficiently solve the location optimization, but also reflects the authentic crowd evacuation process; (3) the dynamic evacuation demands are accurately determined which can satisfy the evacuation demand of residents at all times; and (4) the improved evacuation model simulates the behavior of adults helping the elderly and children, choice behavior for evacuation routes, and spatial heterogeneity of the age structure.

## 2. Materials and Methods

### 2.1. Study Area

The central urban area of Xinyi City, Jiangsu Province, China, which has a total area of 39.36 km$^2$, was selected as the study area for three reasons. First, the central urban area has a high population density, as well as a high construction and development intensity. The population density during peak hours at the northeast corner reaches 37,100 people per square kilometer. The coupling of the high population aggregation and the high spatial density increases the risk during evacuation. Second, the study area has a high seismic risk. The Tanlu fault zone passes through the central urban area of Xinyi City. On 25 July 1668, an 8.5-magnitude earthquake, which occurred in the Tanlu fault zone, struck near Tancheng City, Shangdong Province [63], and the epicenter was only 50 km away from the study area. Third, the study area has a high composite function, and reflects the assembly of several industries such as trade, finance, business, entertainment, and services, thereby attracting a large number of people to gather there. Moreover, the population distribution varies greatly between daytime and nighttime. Figure 1 illustrates location and range of the study area.

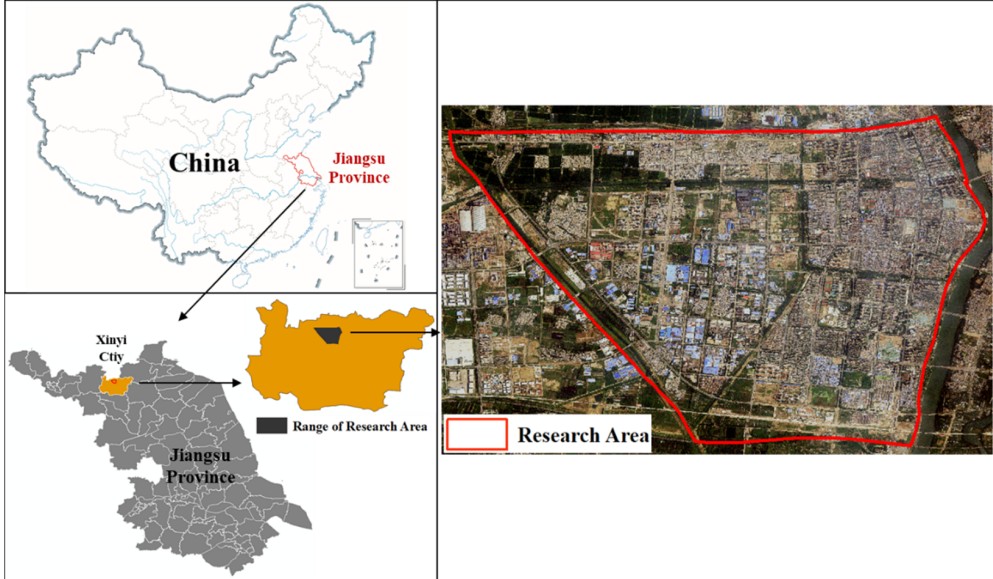

**Figure 1.** The location and range of the study area.

### 2.2. Data Collection and Processing

The optimization model proposed in this study requires the acquisition of spatial data and population data in the study area. The acquired data include the following: (1) the location of refugee demand points and the number of refugees during daytime and nighttime; (2) the number, location, and capacity of shelters that have been built; (3) the number, location, and capacity of candidate shelters; (4) the shortest road network distances from refugee demand points to shelters; and (5) the maximum service distance of shelters. The research objects of this study are resident emergency congregate shelters specified in the relevant Chinese code, so the maximum service distance is 1500 m [1].

The spatial data of the study area include the level of the evacuation roads and the spatial distribution and effective area of current and candidate shelters. The spatial data are derived from OpenStreetMap and Gaode map, and are supplemented and modified via field surveys and remote sensing images (Figure 2). The available land resources in the study area were sorted, and 71 candidate areas that can be used for new shelters are determined. The shortest distances between demand points and shelters along the road network are determined by the Dijkstra algorithm [64], as reported in Table 1.

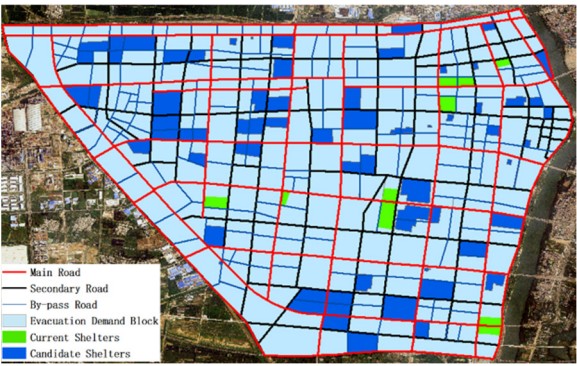

**Figure 2.** The spatial data of the central urban area of Xinyi City.

**Table 1.** The OD matrix of the shortest distances from demand points to resident emergency congregate shelters.

| Demand Points \ Shelters * | 1 | 2 | 3 | ... | 69 | 70 | 71 |
|---|---|---|---|---|---|---|---|
| 1 | 6162 | 893 | 896 | ... | 9692 | 10,006 | 9113 |
| 2 | 5778 | 1486 | 1488 | ... | 9308 | 9621 | 8729 |
| ... | ... | ... | ... | ... | ... | ... | ... |
| 367 | 7504 | 5671 | 5834 | ... | 7365 | 6819 | 5723 |
| 368 | 7273 | 6326 | 6328 | ... | 6698 | 5792 | 5245 |
| 369 | 7547 | 6422 | 6585 | ... | 6972 | 6066 | 5518 |

* Nos. 1–9 are current resident emergency congregate shelters, and Nos. 10–71 are candidate resident emergency congregate shelters.

To obtain high-precision population data, the population in the study area over 24 h was determined based on Chinese mobile phone signaling data from 15 December 2021, to 28 December 2021 (Figure 3). However, these population data do not represent the total number of people, and the proportion of the total population using Chinese mobile communication equipment in the study area is unknown. The nighttime population determined by mobile signaling data was corrected with reference to the permanent population, thus revealing the proportion of the population that uses China's mobile communication equipment that accounts for the permanent population in the study area. The Xinyi Statistical Yearbook provides data on the permanent population. The study area spans the

Xin'an, Beigou, and Mohe neighborhoods. Based on the population data provided by the Xinyi Statistical Yearbook (2020), the numbers of Chinese mobile phone users in the Xin'an, Beigou, and Mohe neighborhoods were, respectively, found to account for 16.67%, 15.46%, and 15.26% of the permanent population of the three districts. The population data obtained from the mobile signaling data were corrected proportionally to determine the evacuation demands at each hour of the day. Finally, the proportion that the number of refugees accounts for the total population in the planning of resident emergency congregate shelters is determined by Equation (1) [1].

$$P_{\text{homeless}} = \frac{1}{A_{per}}\left(\frac{2}{3}A_1 + A_2 + \frac{7}{10}A_3\right) \tag{1}$$

where $P_{\text{homeless}}$ represents the number of homeless people after the earthquake; $A_{\text{per}}$ represents per capita living space of urban residents (m²); $A_1$ represents area of complete damage to residential buildings (m²); $A_2$ represents area of extensive damage to residential buildings (m²); and $A_3$ represents area of moderate damage to residential buildings (m²). Table 2 demonstrates the number of refugees at each moment and the age structure of the demand points.

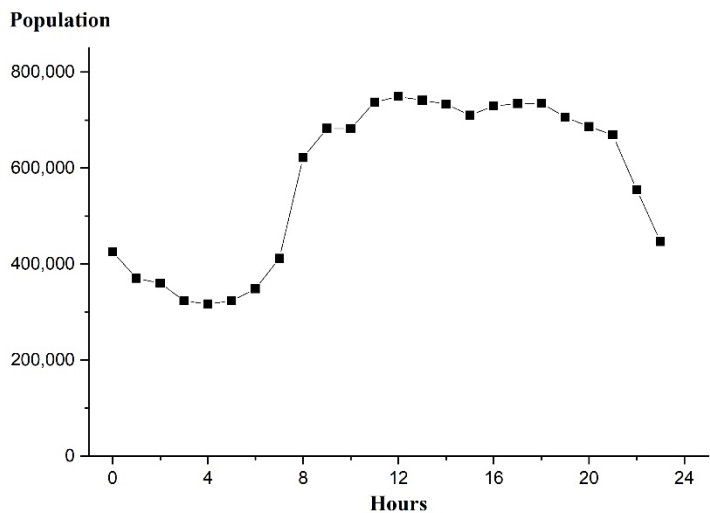

**Figure 3.** Population changes in the study area over 24 h.

**Table 2.** The number of refugees at each moment and the age structure of the demand points.

| Serial Number of Demand Points | Evacuation Demands at Different Times (the Number of People) | | | | | | | Proportion of the Elderly | Proportion of Children |
|---|---|---|---|---|---|---|---|---|---|
| | 0:00 | 1:00 | ... | 12:00 | ... | 22:00 | 23:00 | | |
| 1 | 1122 | 977 | ... | 1974 | ... | 1462 | 1178 | 13.25% | 9.51% |
| 2 | 2543 | 2213 | ... | 4475 | ... | 3313 | 2669 | 13.16% | 8.74% |
| ... | ... | ... | ... | ... | ... | ... | ... | ... | ... |
| 367 | 1420 | 1236 | ... | 2499 | ... | 1850 | 1491 | 13.91% | 10.46% |
| 368 | 701 | 610 | ... | 1233 | ... | 913 | 736 | 14.39% | 9.05% |
| 369 | 471 | 410 | ... | 829 | ... | 614 | 494 | 12.75% | 8.02% |

WorldPop (https://www.worldpop.org/ (accessed on 8th October 2020)) provides grid data for population density at 100 m × 100 m dimensions. Regarding the age structure, WorldPop considers the age segment to be 5 years, and provides the number of people of all age groups in China. The numbers and proportions of elderly people over 65 years old and children under 10 years old at each demand point were determined by superposition analysis in ArcGIS (Table 2).

### 2.3. Construction of the Combined Static/Dynamic Two-Stage Shelter Location Optimization Methodology

2.3.1. Overall Research Framework

Based on the optimization principles of fairness, efficiency, and economy, the number of newly-built shelters and the total evacuation time are taken as the objective functions of the present study. A two-stage shelter optimization methodology based on the combination of static analysis and dynamic simulation is established. Fairness indicates that the optimization results satisfy the evacuation demands of all people. Efficiency represents a rapid evacuation process and the minimization of the evacuation time. Economy refers to minimizing the construction cost, i.e., minimizing the number of newly-built shelters. In the first stage, the static network analysis method is adopted, and the genetic algorithm is designed to determine the minimum number of shelters that can satisfy the evacuation demands of all people under the limitations of the shelter capacity and service distance. In the second stage, the dynamic evacuation models of feasible schemes identified in the first stage are established to determine the optimal scheme with the shortest total evacuation time. The reason for establishing a combined static/dynamic two-stage optimization method is explained as follows. On the one hand, while traditional static network analysis can be solved quickly based on the heuristic intelligent algorithm, it does not take into account the differences in the ages of refugees, the authentic evacuation behavior of pedestrians, and the real congestion state of evacuation road. On the other hand, the dynamic evacuation model based on the improved social force model established in Section 2.3.3 can reflect the selection of evacuation routes, the real-time congestion of evacuation roads, real evacuation behaviors, and age differences. However, if the optimization of shelter location is completed based on a dynamic evacuation simulation, it is necessary to compare the construction cost and evacuation time of each scheme. The number of candidate shelters is $m$, and $i$ is the serial number of the candidate shelter, so the total number of shelters location schemes is determined as $\sum_{i=1}^{m} C_m^i$. $\sum_{i=1}^{m} C_m^i$ types of evacuation simulation models should be established and compared with each other via the exhaustive method, which leads to a large amount of calculation. Therefore, by combining the advantages of the two methods, a two-stage shelter location optimization method based on the combination of static analysis and dynamic simulation is proposed. Specifically, this method includes the following:

(1) Basic data acquisition. The road hierarchy, the spatial location of evacuation demands, and the shelters in the study area are determined from remote sensing images, Google Maps, OpenStreetMap [65], and field research. Some scholars have determined the spatiotemporal distribution of the population based on big data technologies such as point-of-interest (POI) data [66], mobile signaling data [67], Baidu heat maps [68], and social media sign-in data [69]. In the present study, the 24-h population change curve and dynamic evacuation demands in different periods are determined based on mobile phone signaling data. Moreover, the age structures of refugees are determined by data from WorldPop (https://www.worldpop.org/ (accessed on 8th October 2020)).

(2) The optimization in the first stage aims at minimizing the number of shelters via static network analysis. First, the circular evacuation allocation rule is proposed. Circular evacuation allocation indicates that refugees at the demand point are allocated to the surrounding shelters based on the gravity model. If the number of refugees received by one shelter reaches the capacity limitation of the shelter, the evacuation will stop at this time. Pedestrians who have not been evacuated will enter the next cycle until all people have completed the evacuation. Second, under the constraints of the capacity and service distance, a genetic algorithm is designed to solve the minimum number of newly-built shelters. Third, the exhaustive method is adopted to select the feasible schemes that satisfy the dynamic evacuation demands of all 24 h and the limitations of the capacity and service distances for $C_B^k$ (B is the total number of candidate shelters) types of candidate schemes;

(3) Establishment and improvement of the regional emergency evacuation model. Considering the choice of evacuation routes, this study improves the traditional social force model and establishes a selection function for evacuation routes. Furthermore, the spatial

heterogeneity of the age structure of pedestrians and the behavior of adults helping children and the elderly during the evacuation process are considered. The simulation platform Anylogic is used to simulate the evacuation process to determine the total evacuation time and identify the congested road sections in the study area; and

(4) In the second stage, the evacuation time is taken as the objective function for optimization, and the emergency evacuation simulation of the feasible schemes identified in the first stage is completed. The optimal shelter location scheme with the minimum number of shelters and the shortest evacuation time is then determined.

Figure 4 presents the framework of the methodology proposed in this study.

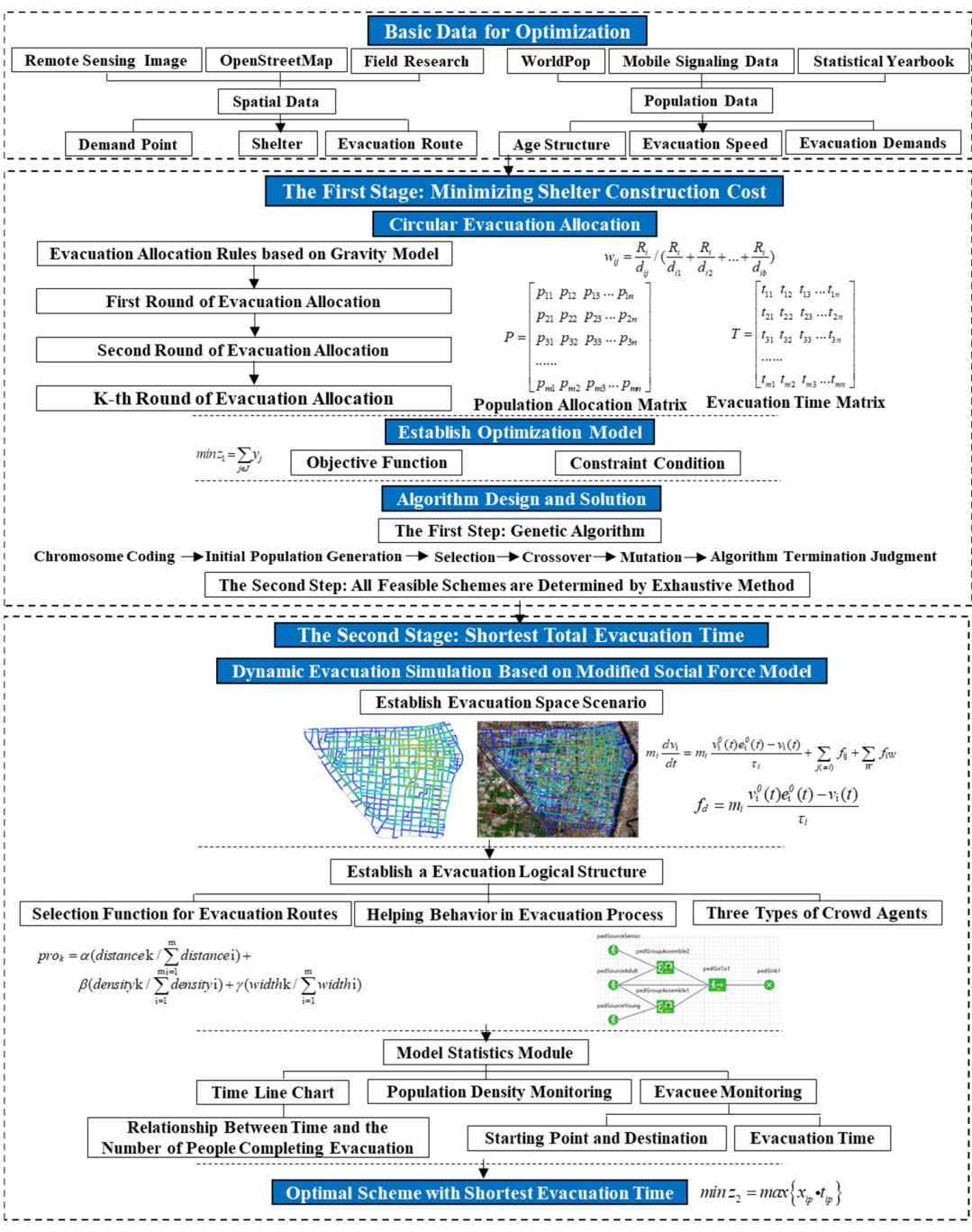

**Figure 4.** The framework of the proposed methodology.

2.3.2. First-Stage Optimization Based on Static Network Analysis: Taking the Construction Cost as the Optimization Objective

1.  Basic assumptions

The optimization model follows four assumptions: (1) The evacuation roads remain safe and reliable after the disaster, regardless of the blocking on evacuation roads; (2) The effective width of the evacuation roads corresponds to their hierarchy. The width of main road, secondary road, and by-pass road are 30 m, 20 m, and 18 m, respectively [70]; (3) In terms of areas occupied by pedestrians, Urban Road Engineering Design Code (CJJ 37-2012) [71] indicates that the width of a group of pedestrians is 0.75 m. Furthermore, when the average interval distance is 1.22–1.34 m, pedestrians can move freely without congestion and trampling [72]. Therefore, in this study, the lateral spacing of pedestrians is set to 0.75 m, and the longitudinal spacing is set to 1.34 m; and (4) Considering the differences in the age structure at different demand points, a simplified method is adopted to calculate the average evacuation speed. It is assumed that all the people at the same demand point go to shelters at the average evacuation speed $v_{ij}$, which is calculated by Equation (2).

$$v_{ij} = (2 \times p_{ic} \times v_c) + ((p_{ia} - p_{ic} - p_{io}) \times v_a) + (2 \times p_{io} \times v_o) \tag{2}$$

where $p_{ic}$, $p_{ia}$, and $p_{io}$ respectively represent the proportions of children, adults, and the elderly at demand point $i$, and $v_c$, $v_a$, and $v_o$, respectively, represent the evacuation speeds of children, adults, and the elderly. In this study, it is assumed that an adult helps a child or a senior citizen to evacuate, and the evacuation speed is determined as the lower value of the evacuation speed between the helper and the assisted. The speeds of children, adults, and the elderly were determined to be 1.05, 1.27, and 1.12 m/s, respectively [73]. The numbers and proportions of children, adults, and the elderly at each demand point were determined from WordPop.

2.  Circular evacuation allocation model

The shelter location optimization problem includes two parts, namely, shelter location and the formulation of the evacuation plan. After determining the locations and construction areas of newly-built shelters, it is necessary to develop an evacuation distribution plan to evacuate refugees to the shelters. The traditional model assumes that refugees at the same demand point are allocated to the same shelter. In this study, the traditional allocation rules are improved and circular evacuation allocation rules are proposed, based on the gravity model. Under the limitations of the service distance and shelter capacity, people at the same demand point can be allocated to several shelters according to the distances from the demand points to the shelters (Figure 5). The location optimization model of shelters established in this study considers the dynamic change of evacuation demands. The evacuation demands within a day are divided into 24 categories, namely, the number of refugees from 0:00 to 24:00. In this section, the evacuation demand at 16:00 is provided as an example to illustrate the circular allocation model.

(a) Evacuation allocation in the first round

In terms of the selection of target shelters for demand points, the population is allocated according to the gravity model. The population at each demand point can be allocated to multiple shelters. The rule for the gravity model is that the number of refugees at one demand point is allocated based on the distances from the demand point to shelters. Shelters closer to the demand point receive more people. If there are $b$ shelters within 1500 m of demand point $i$, the weight coefficient from demand point $i$ to shelter $j$ is determined by Equations (3) and (4). The weight coefficient from demand point $i$ to shelter $j$ is directly proportional to the road network distance from demand point $i$ to shelter $j$.

$$w_{ij} = \frac{1}{d_{ij}} / \left( \frac{1}{d_{i1}} + \frac{1}{d_{i2}} + \ldots + \frac{1}{d_{ib}} \right) \tag{3}$$

$$R_i = d_{i1} + d_{i2} + d_{i3} + \ldots d_{ib} \tag{4}$$

where $d_{ij}$ represents the shortest road network distance from demand point $i$ to shelter $j$. The best evacuation route from each demand point to its assigned shelters is calculated using the Dijkstra algorithm [64]. Then, the number of refugees from demand point $i$ allocated to shelter $j$ is determined by Equation (5):

$$p_{ij} = [w_{ij}S_i] \tag{5}$$

where $w_{ij}$ is the allocation weight, $S_i$ is the total population of demand point $i$, and $p_{ij}$ is the integration function.

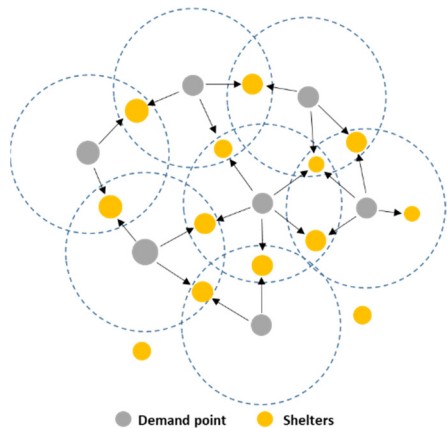

**Figure 5.** The circular evacuation allocation rules from demand points to shelters.

When the number of refugees received by shelter $j$ reaches the maximum capacity of the shelter $v_j$ at moment $t_j^1$, the shelter is closed. At this time, the number of people who have not completed evacuation enters the next cycle. $t_j^1$ is the moment when the number of people received by shelter $j$ reaches the shelter capacity in the first cycle. The population allocation matrix at 16:00 in the first round ($P^1(t = 16)$) can be determined by Equation (5).

$$P^1(t = 16) = \begin{bmatrix} p_{11}^1 & p_{12}^1 & p_{13}^1 \cdots p_{1n}^1 \\ p_{21}^1 & p_{22}^1 & p_{23}^1 \cdots p_{2n}^1 \\ p_{31}^1 & p_{32}^1 & p_{33}^1 \cdots p_{3n}^1 \\ & \cdots & \\ p_{m1}^1 & p_{m2}^1 & p_{m3}^1 \cdots p_{mn}^1 \end{bmatrix}$$

where the matrix columns are the number of people who completed evacuation in the first round at demand points $1, 2, \ldots m$, and the matrix rows are the population received by shelters $1, 2, \ldots n$ in the first round. Moreover, $\sum\limits_{i=1}^{m} p_{ij}^1$ represents the number of people who were received at shelter $j$ in the first round. The value of $\sum\limits_{i=1}^{m} p_{ij}^1$ is determined as the sum of the $j$th column. $\sum\limits_{j=1}^{n} p_{ij}^1$ represents the number of people who completed evacuation in the first round at demand point $i$. The value of $\sum\limits_{j=1}^{n} p_{ij}^1$ is determined as the sum of the $i$th row. The number of people who have completed evacuation in the first round is subtracted from the original total population to determine the population in the next allocation round. The horizontal width and longitudinal spacing between pedestrians on the evacuation road are set as 0.75 m and 1.34 m, respectively. According to the road hierarchy in the study area, the effective width of each evacuation road is determined as $w$ meters. The number of pedestrians accommodated on the lateral space at the same time is $[w/0.75]$.

The time required for the crowd to evacuate from demand point $i$ to shelter $j$ is determined by Equation (6).

$$t_{ij} = (p_{ij}/[w/0.75] \times 1.34 + d_{ij})/v \tag{6}$$

Then, the evacuation time matrix in the first round at 16:00 ($T^1(t = 16)$) can be determined.

$$T^1(t = 16) = \begin{bmatrix} t_{11}^1 & t_{12}^1 & t_{13}^1 \cdots t_{1n}^1 \\ t_{21}^1 & t_{22}^1 & t_{23}^1 \cdots t_{2n}^1 \\ t_{31}^1 & t_{32}^1 & t_{33}^1 \cdots t_{3n}^1 \\ & \cdots & \\ t_{m1}^1 & t_{m2}^1 & t_{m3}^1 \cdots t_{mn}^1 \end{bmatrix}$$

(b) Evacuation allocation in the second round

The number of people who completed the evacuation at demand point $i$ in the first round ($p_i^1$) is subtracted from the total population of demand point $i$ to determine the population in the second allocation round ($p_i^2$). The population distribution matrix and evacuation time matrix of the second round are determined by the same calculation method as that used in the first round.

$$P^2(t = 16) = \begin{bmatrix} p_{11}^2 & p_{12}^2 & p_{13}^2 \cdots p_{1n}^2 \\ p_{21}^2 & p_{22}^2 & p_{23}^2 \cdots p_{2n}^2 \\ p_{31}^2 & p_{32}^2 & p_{33}^2 \cdots p_{3n}^2 \\ & \cdots & \\ p_{m1}^2 & p_{m2}^2 & p_{m3}^2 \cdots p_{mn}^2 \end{bmatrix} \quad T^2(t = 16) = \begin{bmatrix} t_{11}^2 & t_{12}^2 & t_{13}^2 \cdots t_{1n}^2 \\ t_{21}^2 & t_{22}^2 & t_{23}^2 \cdots t_{2n}^2 \\ t_{31}^2 & t_{32}^2 & t_{33}^2 \cdots t_{3n}^2 \\ & \cdots & \\ t_{m1}^2 & t_{m2}^2 & t_{m3}^2 \cdots t_{mn}^2 \end{bmatrix}$$

If the evacuation has not been completed, the third, fourth, etc., rounds of evacuation will be carried out until all people have completed the evacuation.

(c) Total calculation

According to Equation (7), the population allocation matrices of all rounds are added together to determine the total population allocation matrix at 16:00.

$$P(t = 16) = P^1(t = 16) + P^2(t = 16) + P^3(t = 16) + \ldots = \begin{bmatrix} p_{11} & p_{12} & p_{13} & \cdots & p_{1n} \\ p_{21} & p_{22} & p_{23} & \cdots & p_{2n} \\ p_{31} & p_{32} & p_{33} & \cdots & p_{3n} \\ & & \cdots & & \\ p_{m1} & p_{m2} & p_{m3} & \cdots & p_{mn} \end{bmatrix} \tag{7}$$

where $m$ and $n$, respectively, represent the total numbers of demand points and shelters.

Similarly, the total evacuation time matrix at 16:00 is determined by Equation (8).

$$T(t = 16) = T^1(t = 16) + T^2(t = 16) + T^3(t = 16) + \ldots = \begin{bmatrix} t_{11} & t_{12} & t_{13} & \cdots & t_{1n} \\ t_{21} & t_{22} & t_{23} & \cdots & t_{2n} \\ t_{31} & t_{32} & t_{33} & \cdots & t_{3n} \\ & & \cdots & & \\ t_{m1} & t_{m2} & t_{m3} & \cdots & t_{mn} \end{bmatrix} \tag{8}$$

The total time for all people to complete the evacuation can be calculated as the maximum value of the element in matrix $T$ ($t = 16$). The total evacuation distance can be determined by Equation (9). The population $p_{ij}$ at each demand point $i$ and the number of pedestrians received by each shelter $\sum_{i=1}^{m} p_{ij}$ are determined by Equations (7) and (8), respectively. The evacuation time at each demand point is calculated by Equation (10).

$$D = \sum_{i}^{m}\sum_{j}^{n} (p_{ij} \times d_{ij}) \tag{9}$$

$$\text{Evacuation time } (i) = \text{Max } (t_{ij}) \, i = 1, 2, \ldots m; j = 1, 2, \ldots n \tag{10}$$

3. Establishment of the optimization model in the first stage

In this study, equity, efficiency, and the minimization of the construction cost are taken as the principles of shelter location optimization. The number of newly-built shelters and the total evacuation time are taken as objective functions in the optimization model to establish a two-stage shelter location optimization model. In the first stage, under the limitation of the shelter capacity and service distance, the minimum number of shelters to be constructed is determined. The objective function is as follows:

$$min \quad z_1 = \sum_{j \in J} y_j \tag{11}$$

The constraint conditions are as follows:

$$\sum_{j \in J} x_{ij} \geq 1 \ \forall i \in I \tag{12}$$

$$x_{ij} \leq c_{ij} \ \forall i \in I \ \forall j \in J \tag{13}$$

$$x_{ij} \leq y_j \ \forall i \in I \ \forall j \in J \tag{14}$$

$$\sum_{i \in I} [x_{ij} \cdot S_i(t)] \leq y_j \cdot v_j \ \forall j \in J \ t = 1, 2, 3 \dots 23, 24 \tag{15}$$

$$x_{ij} = 0 \ \text{ or } \ 1 \ y_j = 0 \ \text{ or } \ 1 \tag{16}$$

$$y_{1,2,\dots A} = 1 \tag{17}$$

$$\sum_{j=1}^{n} p_{i,j}(t) = S_i(t) \ \forall i \in I \ t = 1, 2, 3 \dots 23, 24 \tag{18}$$

In these equations, $S_i(t)$ represents the number of refugees at demand point $i$ at moment $t$, $v_j$ represents the capacity of shelter $j$, $J$ is the set of shelters, $j \in J$, $I$ is the set of demand points, $i \in I$, and $p_{i,j}(t)$ represents the number of refugees evacuating from demand point $i$ to shelter $j$ at moment $t$. Moreover, $c_{ij}$ is a 0–1 matrix: if the distance from demand point $i$ to shelter $j$ is less than the serviced distance limitation of shelters (1500 m), the value of $c_{ij}$ is determined as 1; otherwise, the value is 0. Furthermore, $x_{ij}$ and $y_j$ are binary decision variables. If shelter $j$ is selected, the value of $y_j$ is determined as 1; otherwise, the value is 0. If demand point $i$ is serviced by shelter $j$, the value of $x_{ij}$ is determined as 1; otherwise, the value is 0. A represents the number of shelters that have been built. The serial number of shelters that have been built are set as 1, 2, . . . ,A. B represents the number of candidate shelters. The serial number of candidate shelters are set as A + 1, A + 2, . . . , A + B.

Equation (11) is the objective function, which represents the minimization of the number of newly-built shelters. Equations (12)–(18) are constraint conditions. Equation (12) ensures that all demand points are covered by at least one shelter. Equation (13) ensures that refugees can only select shelters within the distance limitation (1500 m). Equation (14) ensures that the demand points can only be serviced by the selected shelters. Equation (15) ensures that the population accepted by the shelter will not exceed its capacity at moment $t$. Equation (16) represents the constraints for binary decision variables. Equation (17) ensures that the built shelters are selected (A represents the number of shelters that have been built, while B represents the number of candidate shelters). Finally, Equation (18) ensures that all the refugees complete the evacuation at moment $t$.

4. Design for the solution algorithm

The optimization model in the first stage follows the requirements of the given constraints (the distance limit, capacity limit, and full coverage for refuge demand), which is an NP-hard problem [74]. Therefore, it is difficult to solve the model via the exhaustive method and the mathematical solver Lingo [75]. Thus, in the first stage, the genetic algorithm [76]

is used to solve the model and determine the minimum number of shelters $k$. The process by which the genetic algorithm determines the minimum number of shelters is as follows.

(a) Chromosome coding

The chromosome coding method in this study is binary code. The mapping rule between the location scheme and genetic code is that whether or not a candidate shelter is selected is represented by 0 and 1 in gene encoding. If the candidate shelter is selected, the value of gene encoding is determined as 1. If the candidate shelter is not selected, the value of gene encoding is determined as 0. The length of the chromosome is equal to the number of candidate shelters. Each gene point represents a candidate shelter, and each chromosome represents a shelter location scheme.

(b) Initial population generation

In this study, the random generation method is adopted to generate the initial population. The probability that each candidate shelter is selected is determined as 50%, thus generating a series of initial chromosomes. If the size of the initial population is small, the algorithm is prone to a local extreme point. Otherwise, if the size of the initial population is large, the number of iterations in the evolution process will greatly increase, thereby affecting the efficiency of the algorithm. To avoid these two extreme situations, the population size is determined as 100 in this study.

(c) Determining the fitness function value of each location scheme

The fitness function value is the standard for selecting chromosomes (i.e., the shelter location scheme) and judging the merits of chromosomes. In the first stage, the fitness function is determined as the number of shelters. The penalty function is introduced into the calculation for the fitness function value. If a shelter location scheme does not satisfy the constraint conditions, the number of shelters of the scheme plus 100 is its fitness function value.

(d) Selection operation

Roulette wheel selection and the elitism strategy are adopted to complete the selection operation in this study. First, the fitness function value $f_k$ of each chromosome in the population is calculated. The probability that each chromosome is selected is proportional to its fitness function value. The selective probability is determined by Equation (19) [76].

$$P_c = f_k / \sum_{i=1}^{100} f_i \tag{19}$$

Moreover, the elite retention strategy is also adopted. The chromosome in the population with the best fitness function value is directly copied into the next generation without matching and crossover, which can ensure that the optimal fitness value of the generation is not less than that of the previous generation.

(e) Crossover operation

The hybrid crossover strategy of multi-point and single-point crossover is adopted in this study. At the early stage of evolution, multi-point crossover can maintain the diversity of solutions and enhance the exploration ability of the algorithm. In the later stage, single-point crossover can accelerate the convergence of the algorithm and improve the computation speed. Crossover operation is adopted with fixed probability $P_c$. First, 10 crossover points are randomly set in the parent chromosomes to exchange gene blocks. Then, single-point crossover is adopted to determine the crossover position of two chromosomes generated by multi-point crossover, and the progeny chromosomes are obtained.

(f) Mutation operation

In the crossover algorithm, the progeny chromosomes not only inherit the information of the parent chromosomes, but also mutate with a certain probability $p_m$. The mutation strategy adopted in this study is that a random number $r_d \in (0,1)$ is first generated for a gene point. If the value of $r_d$ is less than $p_m$, the value of the gene point is changed. The mutation operation is completed for each gene point until a new chromosome is generated.

(g) Algorithm termination judgment

Considering the convergence time and accuracy, the iterative termination conditions are determined as follows: (1) the optimal fitness function solution does not improve in successive generations; and (2) maximal genetic evolution algebra. The gene points of the optimal chromosome correspond to the optimal shelter location optimization scheme. The fitness function value of the optimal chromosome is the minimum number of shelters.

After determining the minimum number of newly-built shelters via the genetic algorithm, the schemes that satisfy the 24-h dynamic evacuation demand under the limitations of the shelter capacity and service distance are selected by the exhaustive method as the feasible location schemes from $C_B^k$ types of candidate schemes.

2.3.3. Second-Stage Optimization Based on Dynamic Evacuation Simulation: Taking the Evacuation Time as the Optimization Objective

1.   Regional evacuation simulation based on the social force model

The social force model was proposed by Helbing et al. [77] and integrated into the underlying algorithm in the Anylogic simulation platform. The social force model has high accuracy and strong continuity, and can simulate the specific evacuation behavior of pedestrians. Moreover, it can be combined with the multi-agent model [55] to simulate the heterogeneity of different types of pedestrians. The social force model assumes that the evacuation process is determined by the following three types of forces: (1) the self-driving force of pedestrians, which represents that human agents actively drive themselves to evacuate; (2) the interaction force between pedestrians, which indicates that human agents repel each other to maintain a certain distance; and (3) the force between pedestrians and the environment, which reflects the repulsive effect between human agents and the environment, such as boundaries and obstacles. The basic principles of the social force model are given by Equations (20) and (21) [75].

$$f_d = m_i \frac{v_i^0(t)e_i^0(t) - v_i(t)}{\tau_i} \tag{20}$$

$$m_i \frac{dv_i}{dt} = m_i \frac{v_i^0(t)e_i^0(t) - v_i(t)}{\tau_i} + \sum_{j(\neq i)} f_{ij} + \sum_W f_{iW} \tag{21}$$

where $f_d$ represents the self-driving force of pedestrians, $f_{ij}$ represents the interaction between pedestrians, $f_{iw}$ represents the interaction between pedestrians and obstacles, mi represents the weight of pedestrians, $v_i^0(t)$ indicates the expected speed, $\tau_i$ represents the acceleration time, and $v_i(t)$ represents the actual evacuation speed.

Research on crowd evacuation behavior shows that when a disaster occurs, the choice for evacuation routes is influenced by the distances from demand points to shelters, the population density, and the width of evacuation roads [78]. In this study, the Java language is used to establish a selection function for evacuation routes on the Anylogic platform. Equation (22) determines the selection probability of evacuation road $k$, and $\alpha$, $\beta$, and $\gamma$ represent the weight coefficients. Moreover, *distance* k, *density* k, and *width* k, respectively, represent the distance following route $k$, the crowd density, and the width of evacuation route $k$.

$$pro_k = \alpha(distance\ k / \sum_{i=1}^{m} distance\ i) + \beta(density\ k / \sum_{i=1}^{m} density\ i) + \gamma(width\ k / \sum_{i=1}^{m} width\ i) \tag{22}$$

The weight coefficients $\alpha$, $\beta$, and $\gamma$ are determined by a video that recorded an authentic evacuation process during an earthquake (https://m.v.qq.com/z/msite/play-short/index.html?cid=&vid=o08073ef0ll&qqVersion=0 (accessed on 24th October 2020)). The evacuation model of a double-exit room was established according to the video of the authentic evacuation. The video records the evacuation process in the real disaster. Using this authentic video to determine $\alpha$ is reliable. The familiarity of the crowd in the classroom with the exit and path is consistent with the familiarity of the crowd with the evacuation

road and destination at neighborhood scale. When the weight coefficients $\alpha$, $\beta$ and $\gamma$ were, respectively, 0.56, 0.24, and 0.2, the simulation result fits well with the authentic scenario (Figure 6).

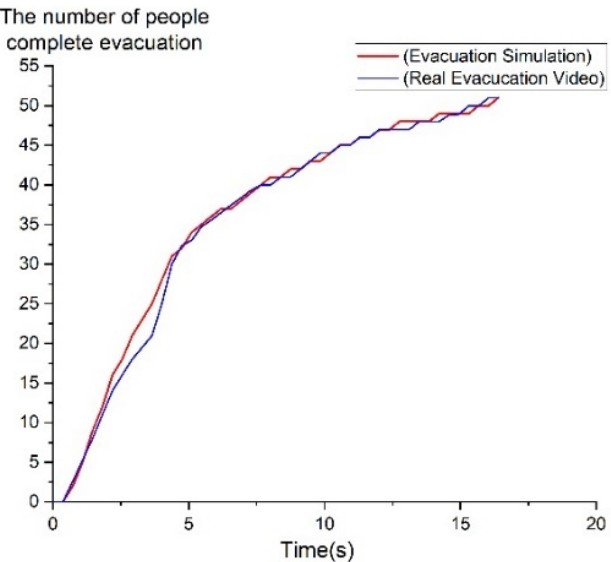

**Figure 6.** The relationship between time and the number of people completing evacuation.

The establishment of a regional evacuation model on the Anylogic platform includes four steps, namely, the construction of an evacuation space scenario, the establishment of a logical structure, the setting of simulation parameters, and the establishment of statistical modules (Figure 4). First, the evacuation space of the study area is established based on both population data and spatial data. Second, due to the different evacuation velocities of people of different ages, the differences in the age structures in the logical structure are considered, and three types of pedestrian agents are constructed, namely the elderly, adults, and children. The expected velocities of pedestrians of different ages are reported in Table 3. In the evacuation process, with the increase of the crowd density, the evacuation velocity will decrease and the movement direction will continue to change. The time-varying evacuation velocities and directions are determined by the social force model embedded in the Anylogic platform [79]. Consistent with the assumption in the first stage, it is also assumed that an adult helps a child or an elderly person evacuate in the evacuation simulation model. The evacuation velocity of aggregation is determined as the lower value of the evacuation speed between the helper and the helped. Third, the time-varying crowd density on the evacuation road, the evacuation time, the starting point and destination of each pedestrian, and the relationship between time and the number of people completing evacuation are determined based on the statistical model.

**Table 3.** The evacuation velocity of people of different ages [62,73].

| Age | Velocity Range (m/s) | Average Velocity (m/s) |
| --- | --- | --- |
| Children | (0.49, 1.63) | 1.05 |
| Adults | (0.65, 1.98) | 1.27 |
| Elderly | (0.58, 1.55) | 1.12 |

2. The implementation process for optimization in the second stage

In the second stage, the improved social force model is established to simulate the evacuation process of pedestrians. The regional dynamic evacuation model considers the selection behavior for evacuation routes and the differences in the age structure of the crowd, and, simultaneously, sets capacity and distance constraints. Based on the evacuation

simulation results, the total evacuation time is taken as the objective function. The shelter location scheme with the shortest evacuation time is finally selected.

In the first stage, the minimum number of newly-built shelters is determined as $k$. In the second stage, the exhaustive method is used to select the schemes that satisfy the dynamic evacuation demands during 24 h and the limitations of the shelter capacity and service distance from $C_B^k$ types of schemes (B is the number of candidate shelters) as the feasible schemes. The objective function of the second stage is demonstrated in Equation (23). The constraints are consistent with those in the first stage. After comparing the overall evacuation time of the feasible schemes identified in the first stage, the optimal scheme with the minimum total evacuation time is selected.

$$min \ z_2 = max\{x_{ip} \cdot t_{ip}\} \tag{23}$$

where $P$ is the set of selected shelters on feasible scheme, $p \in P$. $I$ is the set of demand points, $i \in I$. $t_{ip}$ represents the evacuation time from demand point $i$ to selected shelter $p$.

## 3. Results and Analysis

### 3.1. Optimization Results of Shelter Location Based on the Combination of Static Analysis and Dynamic Simulation

Based on the principles of equity, economy, and efficiency, a two-stage shelter optimization model was established. In the first stage, under the limitations of the capacity and service distance of shelters, the minimum number of newly-built shelters that can satisfy the evacuation demand was determined. According to the designed genetic algorithm, 100 groups of location schemes were randomly established as the initial population. The crossover probability and the mutation probability in the genetic algorithm were determined as 0.9 and 0.3, respectively. The relationship between the fitness function value and the number of iterations is presented in Figure 7. After a trial calculation, when the algorithm was iterated to 200 rounds, the fitness function value was found to reach the minimum value and did not change. The fitness function value was found to converge to 25, indicating that it is necessary to build at least 25 new shelters to satisfy the dynamic evacuation demands during the 24 h of a day. The traditional shelter optimization model does not consider the situation in which people from one demand point have access to multiple shelters, and it is assumed that people at the same demand point enter the same shelter. By comparing the proposed location optimization model based on circular evacuation allocation with the traditional model, the number of newly built shelters was found to be reduced from 42 to 25, which will reduce the construction cost of shelters by 40.4% and improve the shelter utilization efficiency.

In the second stage, under the condition that the number of newly built shelters is determined, the feasible location schemes were selected from $C_{62}^{25}$ schemes by the exhaustive method (Table 4). The feasible schemes should satisfy the limitations of the shelter capacity and service distance, as well as the dynamic evacuation demands during the 24 h of a day. Based on the improved social force model described in Section 2, the regional evacuation models of 15 feasible schemes in Table 4 were established, as presented in Figure 8. Figure 8 demonstrates the evacuation process of pedestrians of Scheme 14 from Table 4. The graphic pattern in lower left is the population density. Change of color represent the dynamically changing population density on the evacuation roads. The total evacuation times of the 15 feasible schemes in Table 4 were compared with each other, and the optimal shelter location scheme with the shortest evacuation time was, ultimately, determined. Table 5 demonstrates the evacuation time of 15 candidate feasible schemes. The spatial distribution and effective areas of the selected emergency shelters are exhibited in Figure 9. There were four shelters with effective areas between 10,000 and 15,000 m$^2$, which accounted for 16% of the total shelters. The numbers of shelters with effective areas between 15,000 and 20,000 m$^2$ and between 20,000 and 50,000 m$^2$ were, respectively, found to be 5 and 11. Moreover, there were 5 shelters with effective areas exceeding 50,000 m$^2$, which account for 20% of the total shelters. Based on the circular evacuation allocation rules formulated in Section 2.3.2,

the evacuation population allocation matrix and the evacuation time matrix, as well as target shelters for residents at each demand point, were determined. Figure 10 illustrates the evacuation directions in the emergency evacuation scenario, which also indicate the connections between demand points and the shelters that people choose to enter. The average evacuation time was found to be about 818 s, and the shortest evacuation time was found to be less than 2 min. The evacuation time of all people was found to be less than 30 min. Figure 11 demonstrates the frequency distribution of the evacuation time; 57.2% of the crowd completed the evacuation in less than 15 min, and 88.5% of the crowd completed the evacuation in less than 20 min. The evacuation times of five demand points were found to be more than 25 min. Compared with the evacuation time before optimization, the average evacuation time for pedestrians from each demand point decreased from 4822 s to 818 s. The evacuation efficiency increased by 83%, and the dispersion of the evacuation time at different demand points decreased significantly (Figure 12).

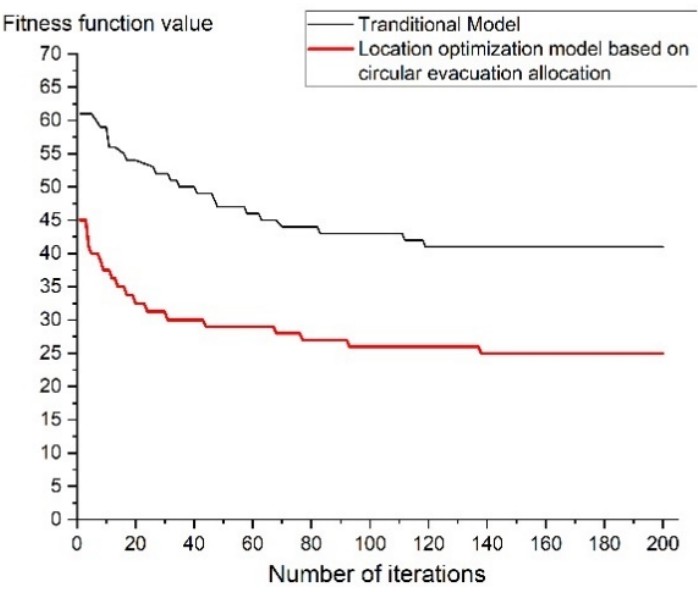

**Figure 7.** The relationship between the fitness function value and the number of iterations.

**Table 4.** The shelter location schemes in the first stage.

| Serial Number of Candidate Shelters | Scheme 1 | Scheme 2 | Scheme 3 | Scheme 4 | ... | Scheme 14 | Scheme 15 |
|---|---|---|---|---|---|---|---|
| 1 | 1 | 1 | 1 | 1 | ... | 1 | 1 |
| 2 | 1 | 1 | 1 | 1 | ... | 1 | 1 |
| ... | ... | ... | ... | ... | ... | ... | ... |
| 8 | 1 | 1 | 1 | 1 | ... | 1 | 1 |
| 9 | 1 | 1 | 1 | 1 | ... | 1 | 1 |
| 10 | 1 | 0 | 1 | 0 | ... | 0 | 0 |
| 11 | 0 | 1 | 1 | 1 | ... | 1 | 1 |
| 12 | 0 | 0 | 0 | 0 | ... | 0 | 0 |
| ... | ... | ... | ... | ... | ... | ... | ... |
| 300 | 0 | 1 | 0 | 0 | ... | 1 | 1 |
| ... | ... | ... | ... | ... | ... | ... | ... |
| 330 | 1 | 1 | 0 | 1 | ... | 0 | 1 |
| ... | ... | ... | ... | ... | ... | ... | ... |
| 369 | 0 | 0 | 0 | 0 | ... | 0 | 0 |

Note: The numbers in the table are the values of $y_i$; 1 represents that the candidate shelter is selected, and 0 represents that the candidate shelter is not selected. Nos. 1–9 are current resident emergency congregate shelters, and Nos. 10–71 are candidate resident emergency congregate shelters.

**Figure 8.** The evacuation model of the study area. (**a**) 5 min, (**b**) 10 min, (**c**) 15 min, and (**d**) 25 min.

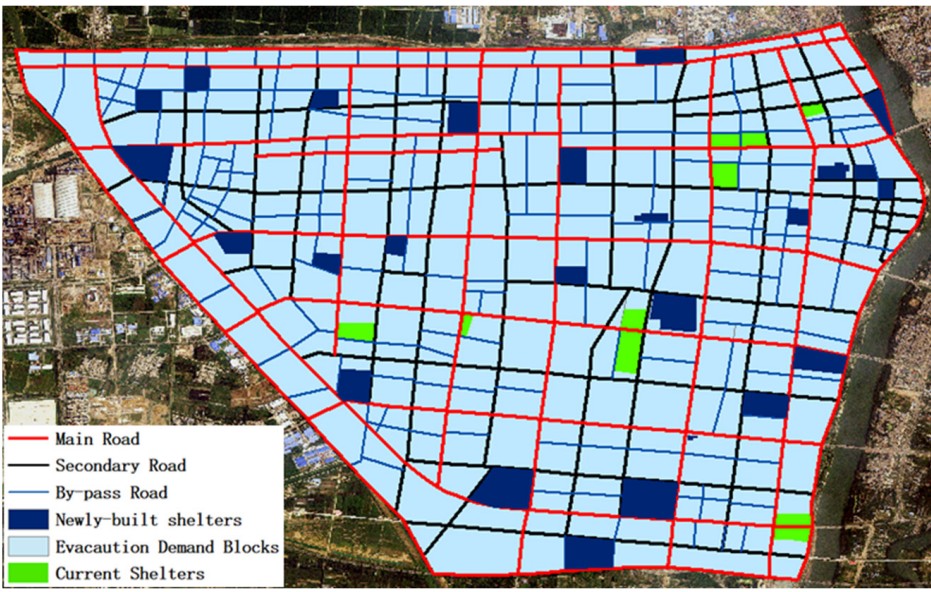

**Figure 9.** The optimal shelter location scheme.

**Table 5.** The evacuation time of 15 feasible schemes.

| | Serial Number of Feasible Schemes | | | | | | | | | | | | | | |
|---|---|---|---|---|---|---|---|---|---|---|---|---|---|---|---|
| | **1** | **2** | **3** | **4** | **5** | **6** | **7** | **8** | **9** | **10** | **11** | **12** | **13** | **14** | **15** |
| Evacuation Time(s) | 2031 s | 2408 s | 1930 s | 1983 s | 2307 s | 1872 s | 2249 s | 1856 s | 2308 s | 1986 s | 1875 s | 2179 s | 2165 s | 1796 s | 2073 s |

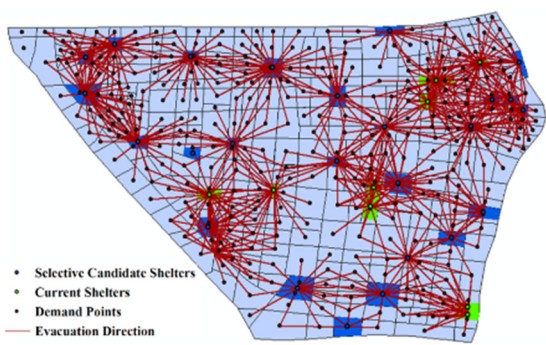

**Figure 10.** The optimal shelter location scheme.

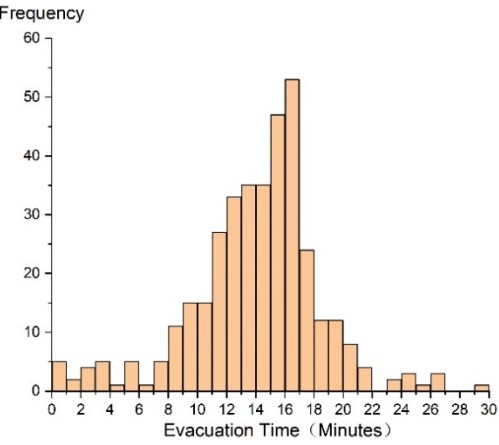

**Figure 11.** The optimal shelter location scheme.

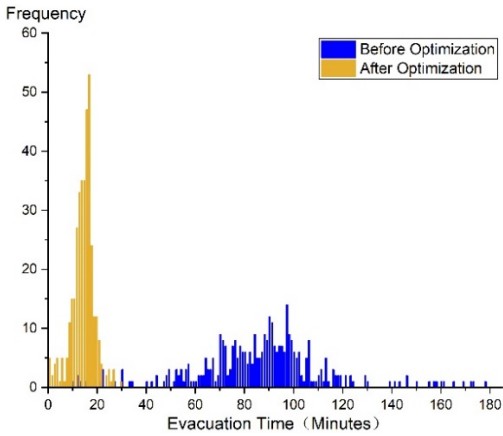

**Figure 12.** The comparison of the frequency distributions of the evacuation time before and after optimization.

### 3.2. Verification by Dynamic Evacuation Simulation

Based on the regional evacuation simulation method described in Section 2.3.3, spatial modeling and parameter setting were completed on the Anylgoic platform, and a Java

program was written to formulate the selection rules for evacuation routes. The evacuation simulation of the study area was completed based on the method presented in Section 2, and the variation of the number of evacuated people arriving at emergency shelters with time was statistically analyzed (Figure 13). After the optimization, the evacuation efficiency of the study area was found to be significantly improved, and the evacuation time was significantly reduced. It took only 1802 s for all people to complete the evacuation, which is 16.8% of the total evacuation time before the optimization. The evacuation efficiency was, therefore, improved by 83.2%. When the evacuation time was 5 min, 43.1% of the pedestrians had reached shelters, which is 3.2 times the number of people who had reached shelters before optimization. Moreover, 95% of the pedestrians had completed the evacuation at 1552 s, which is far shorter than the time required before optimization (5513 s) (Figure 13).

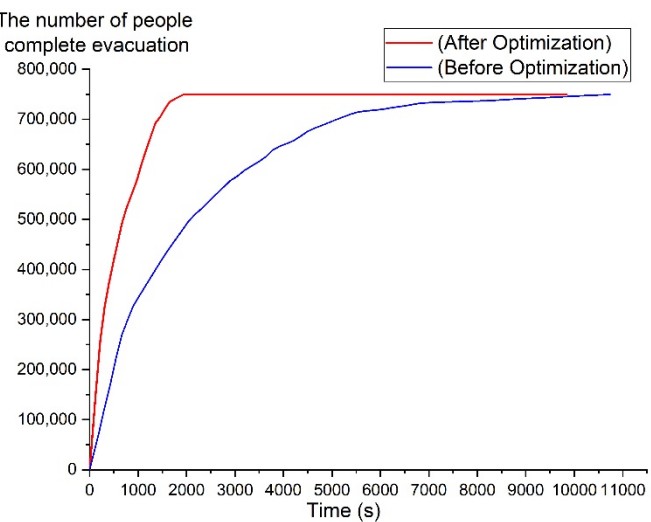

**Figure 13.** The comparison of the numbers of people completing evacuation before and after optimization.

## 4. Discussion

This study established a systematic methodology for the location optimization of shelters. This methodology comprehensively considers the time-varying changes of the congestion state on evacuation roads, the crowd evacuation behavior, and pedestrian psychology via regional evacuation simulation. The traditional static network model can be solved quickly, and the dynamic evacuation simulation can veritably reflect the crowd evacuation process. This study combined the advantages of static network analysis and dynamic evacuation simulation to propose a combined static/dynamic optimization method. In the first stage, from the perspective of government managers, the optimization objective is determined as the minimization of the construction cost of shelters. In the second stage, from the perspective of refugees, the optimal scheme with the shortest overall evacuation time is determined via the dynamic simulation of the evacuation process. This method efficiently, accurately, and scientifically completes the location optimization of shelters. The shelter location optimization methodology based on the combination of static analysis and dynamic simulation breaks through the assumption in the traditional methodology, namely, that pedestrians evacuate following the appointed routes, and considers the real evacuation process as the reference system to accurately simulate the choice of evacuation routes, the differences in the evacuation speed caused by different ages, and the behavior of pedestrians helping others. Compared with the traditional optimization method, the proposed methodology is characterized by the following four advantages:

(1) The deficiencies in the configuration of urban shelters can be intuitively found via emergency evacuation simulation. Furthermore, the overall evacuation time, the number of people accommodated in each shelter, and the evacuation time, evacuation route, and

final target shelter of each pedestrian are determined, thereby better guiding the spatial location and capacity configuration of shelters;

(2) The combined static/dynamic optimization methodology of shelters is proposed. The traditional static network model can be solved quickly, and the dynamic evacuation simulation can truly reflect the crowd evacuation process. Therefore, the proposed method combines the advantages of static network analysis and dynamic evacuation simulation, and considers the number of shelters and the overall evacuation time as the objective functions. The novel methodology cannot only efficiently solve the location optimization, but also reflects the authentic crowd evacuation process;

(3) Dynamic evacuation demands are calculated. In the traditional method, evacuation demands are calculated based on the permanent population, which cannot reflect the spatial and temporal changes of the population and satisfy evacuation demands during the peak period. The proposed method determines the temporal and spatial distributions of the population based on mobile phone signaling data. Therefore, the dynamic evacuation demands are accurately calculated to satisfy the evacuation demand of residents at all times and in all regions; and

(4) A regional evacuation model considering the crowd evacuation behavior and spatial heterogeneity of the age structure was proposed. Via the secondary development of Java programming on the Anylogic platform, the traditional social force evacuation model is improved. The improved evacuation model establishes the selection function for evacuation routes and considers the spatial heterogeneity of the evacuation velocity caused by different ages of pedestrians. Moreover, the behavior of adults helping the elderly and children during an evacuation is simulated.

The combined static/dynamic two-stage optimization methodology proposed in this study can not only be used in central urban area, but can also be used in larger areas, for example, the whole Xinyi city, or other megalopolis in China. The empirical study on larger areas will be conducted in further research. In this study, we choose the central area of Xinyi city because of its high population density, high seismic risk, and high composite function. But the proposed methodology is effective and valid for other areas.

Although this study has made four contributions, the proposed model still has some deficiencies: (1) People's preferences for different types of shelters is not considered in evacuation model. What's more, when a dangerous event occurs, in evacuation conditions, demand models specified and calibrated in ordinary conditions cannot be directly applied due to multiplicity of decision makers [80]. The heterogeneity of evacuation behavior is not considered; and (2) All evacuation roads are assumed to be unobstructed, which is inconsistent with actual situation. After an earthquake occurs, some evacuation routes will be blocked by the debris of buildings.

In addition, determining how to simulate people's preferences for different shelters and how to consider the influence of the disaster environment on the evacuation process will be the focus of future research. We can bring the risk of road blocking and building damage caused by hazards into the evacuation scenario by combining the advanced disaster simulation technology with regional evacuation simulation, so as to determine the evacuation demands more accurately and reflect the evacuation process more authentically. The scientism and accuracy of the evacuation process and the shelter location optimization will be improved.

## 5. Conclusions

This study introduced the methodology and mathematical model in the field of crowd emergency evacuation to the location optimization of shelters. Via the combination of the advantages of static network analysis and dynamic evacuation simulation, a shelter location optimization method was proposed. The novel methodology can not only efficiently solve the location optimization, but also reflects the authentic crowd evacuation process. The crowd evacuation behavior and spatial heterogeneity of the age structure are considered in evacuation model. Mean-while, the dynamic evacuation demands were accurately

calculated to satisfy the evacuation demands at all times. Taking the central urban area of Xinyi City, Jiangsu Province, China, as an example, the feasibility and effectiveness of this methodology were verified. The following conclusions were obtained:

(1) The first stage aims to minimize the number of shelters. When the number of iterations of the genetic algorithm reached 200 generations, the fitness function value converged to 25. Thus, to satisfy the dynamic evacuation demands of all residents, 25 new shelters are required. Compared with the traditional optimization model assuming that pedestrians at the same demand point enter the same shelter, the proposed location optimization model based on circular evacuation allocation rules was found to reduce the number of newly-built shelters from 42 to 25, thus reducing the construction cost by 40.4% and improving the utilization efficiency of shelter space resources; and

(2) A total of 25 shelters that satisfied the dynamic evacuation demands and the limitations of the service distance and capacity were selected from 71 candidate shelters in the first optimization stage, and 15 feasible schemes were determined. In the second stage, the total evacuation time was taken as the optimization object, and the total evacuation times of the 15 feasible schemes were compared to determine the optimal location scheme. After optimization, the evacuation time was found to be greatly reduced to 1802 s, and the evacuation efficiency was increased by 83.2%. The average evacuation time of all pedestrians was found to be 818 s, the shortest evacuation time was within 2 min, and the longest evacuation time was less than 30 min. When the evacuation time reached 5 min, 43.1% of the pedestrians had reached shelters, which is 3.2 times the number of pedestrians that had reached shelters before optimization. Moreover, 95% of the pedestrians were found to have completed evacuation at 1552 s, which is far shorter than that before optimization (5513 s). Compared with the evacuation time before optimization, the average evacuation time for pedestrians from each demand point decreased from 4822 s to 818 s. Moreover, the evacuation efficiency increased by 83%, and the dispersion of the evacuation time at different demand points decreased significantly.

**Author Contributions:** Conceptualization, Guangchun Zhong and Guofang Zhai; Data curation, Guangchun Zhong and Wei Chen; Methodology, Guangchun Zhong and Guofang Zhai; Programming, Guangchun Zhong and Wei Chen; Writing—original draft preparation, Guangchun Zhong; Writing—review and editing, Guofang Zhai and Wei Chen; Visualization, Guofang Zhai and Guangchun Zhong; Supervision, Guofang Zhai and Wei Chen; Project administration, Guofang Zhai; Funding acquisition, Wei Chen. All authors have read and agreed to the published version of the manuscript.

**Funding:** This research was supported by the National Natural Science Foundation of China [Grant No. 52108053]; the Natural Science Foundation of Jiangsu Province [Grant No. BK20200762]; the Social Science Foundation of Jiangsu Province [Grant No. 20ZZC001]; the University Social Science Research Project of Jiangsu Province [Grant No. 2020SJA0098]; and the University Natural Science Research Project of Jiangsu Province [Grant No. 20KJB560024]).

**Institutional Review Board Statement:** Not applicable.

**Informed Consent Statement:** Not applicable.

**Data Availability Statement:** Not applicable.

**Acknowledgments:** We thank the editor and anonymous referees who read the paper and provided helpful comments for improvements.

**Conflicts of Interest:** The authors declare no conflict of interest.

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
