# Peer review of "Optimization of Shelter Location Based on a Combined Static/Dynamic Two-Stage Optimization Methodology: A Case Study in the Central Urban Area of Xinyi City, China"

_ijgi, doi:10.3390/ijgi11040262_

Round 1

Reviewer 1 Report

  1. The English tense of some sentences needs further correction. For example, lines 91 and 92, 201-203, etc.
  2. What method does "the exhaustive method" refer to in detail?
  3. In line 354, it is not necessary for the parameter Ri to exist in Formula (3).
  4. Line 375,“The horizontal spacing and longitudinal spacing” should be “The horizontal width and longitudinal spacing”? Conflict with line 315.
  5. Line 607, it's strange that "Chapter 3" appears.
  6. How to calculate the last column in the first row of Table 4? In addition, how to determine the 15 schemes that need to be processed in the second stage is not explained clearly.
  7. Description is inaccurate and clear in Figure 8. For example, what does the change of color represent?
  8. The optimization of the two stages in this paper is relatively independent. How to balance the evacuation time and construction cost is not considered in this paper. In addition, the design of the capacity of each shelter is also considered.
  9. Only considering the central urban area is not perfect. For example, people work in the central area of the city during the day and live in the suburbs at night. As a result, the population distribution during the day and at night has obvious temporal and spatial differences. Therefore, it is suggested to consider the optimal location of shelters from a larger area.
  10. It is suggested that the author compare the optimization results with other existing shelter location schemes, rather than a simple comparison before and after optimization based on the proposed in this paper, because even if it is not optimized, the evacuation efficiency will be improved to a certain extent by randomly adding shelters.
  11. It is suggested that the author show the comparison of evacuation time of 15 schemes, so as to see the difference of these results in evacuation efficiency to a certain extent.

Author Response

Response to Reviewer 1 Comments

Point 1: The English tense of some sentences needs further correction. For example, lines 91 and 92, 201-203, etc.

Response 1: The English tense from lines 91 to lines 105, from lines 198 to lines 204 are corrected in the revised manuscript.

Point 2: What method does "the exhaustive method" refer to in detail?

Response 2: "The exhaustive method" means enumeration for every situation. In the first optimization stage in this paper, the minimum number of newly-built shelters are determined via the genetic algorithm. B is the totoal number of candidate shelters, k is the the minimum number of newly-built shelters determined in the first stage. The number of candidate location schemes is Ck B. Then conducting trial on each candidate location scheme whether it can satisfy the 24-h dynamic evacuation demand. Finally, the schemes which satisfy the 24-h dynamic evacuation demand under the limitations of the shelter capacity and service distance are selected by exhaustion from Ck B types of candidate schemes.

Point 3: In line 354, it is not necessary for the parameter Ri to exist in Formula (3).

Response 3: Yes, it is not necessary for the parameter Ri to exist in Formula (3). Formula (3) is modified as follows in the revised manuscript:

wij=1/dij(1/di1+1/di1+...+1/dib)

Point 4: Line 375,"The horizontal spacing and longitudinal spacing" should be "The horizontal width and longitudinal spacing"? Conflict with line 315.

Response 4: Yes, "The horizontal spacing and longitudinal spacing" is revised as "The horizontal width and longitudinal spacing" in the revised manuscript.

Point 5: Line 607, it's strange that "Chapter 3" appears.

Response 5: Line 607, "According to the genetic algorithm designed in Chapter 3" is revised as "According to the designed genetic algorithm designed" in the revised manuscript.

Point 6: How to calculate the last column in the first row of Table 4? In addition, how to determine the 15 schemes that need to be processed in the second stage is not explained clearly.

Response 6: The result of table 4 is determind by the designed genetic algorithm and the exhaustive method. B is the totoal number of candidate shelters, k is the the minimum number of newly-built shelters which is determined by desinged genetic algorithm from Lines 439 to Lines 513 in the first optimization stage. After the minimum number of newly-built shelters are determiend, the number of candidate location schemes is Ck B. Then conducting trial on each candidate location scheme whether it can satisfy the 24-h dynamic evacuation demand. Finally, the schemes which satisfy the 24-h dynamic evacuation demand under the limitations of the shelter capacity and service distance are selected by exhaustion from Ck Btypes of candidate schemes. The selected feasible location schemes are demonstrated in figure 4.

Point 7: Description is inaccurate and clear in Figure 8. For example, what does the change of color represent?

Response 7: The detailed description is added in the revised manuscript. Change of color represent the dynamically changing population density on the evacuation roads. The graphic pattern in lower left is the population density.

The added description is as follows: “Figure 8 demonstrates the evacuation process of pedestrians of scheme 14 from table 4. The graphic pattern in lower left is the population density. Change of color represent the dynamically changing population density on the evacuation roads.”

Point 8: The optimization of the two stages in this paper is relatively independent. How to balance the evacuation time and construction cost is not considered in this paper. In addition, the design of the capacity of each shelter is also considered.

Response 8: This paper proposed a static/dynamic two-stage optimization methodology for optimization location of shelters. Both evacuation time and construction cost are objective functions in the proposed model. The first stage and second stage are tightly interlocked. The result of first stage is the import and basic data for second stage. The first optimization stage determines the minimum number of newly-built shelters. B is the totoal number of candidate shelters, k is the the minimum number of newly-built shelters determined in the first stage. The second optimizatioin stage has two process: (1) conducting trial on each candidate location schemes with minimum number of newly-built shelters. The number of candidate location schemes is Ck B. Then, the schemes which satisfy the 24-h dynamic evacuation demand under the limitations of the shelter capacity and service distance are selected by exhaustion from types of Ck Bcandidate schemes. The selected feasible location schemes are demonstrated in figure 4; (2) completing dynamic evacuation simulation for 15 feasible schemes in figure 4, and the optimal scheme with minimum evacuation time is determined. The construction cost and evacuation time of the optimal scheme are minimal. In short, first stage determines the minimum number of newly-built shelters; the second stage takes the evacuation time as objective function to select the optimal location scheme from candidate location schemes with minimum number of newly-built shelters. The result of first stage is the import and basic data for second stage.

Point 9: Only considering the central urban area is not perfect. For example, people work in the central area of the city during the day and live in the suburbs at night. As a result, the population distribution during the day and at night has obvious temporal and spatial differences. Therefore, it is suggested to consider the optimal location of shelters from a larger area.

Response 9: The combined static/dynamic two-stage optimization methodology proposed in this study can not only be used in central urban area, but can also be used in larger area, for example, the whole Xinyi city, or other megalopolis in China. The empirical study on larger areas will be conducted in further research. In this study, we choose central area of Xinyi city because of its high population density, high seismic risk and high composite function. But the proposed methodology  is effective and valid for other areas.

The above response has been added on Discussion in the revised manuscript.

Point 10: It is suggested that the author compare the optimization results with other existing shelter location schemes, rather than a simple comparison before and after optimization based on the proposed in this paper, because even if it is not optimized, the evacuation efficiency will be improved to a certain extent by randomly adding shelters.

Response 10: Figure 7 compares the proposed location optimization model based on circular evacuation allocation with the traditional shelter location model. The number of newly built shelters was found to be reduced from 42 to 25, which will reduce the construction cost of shelters by 40.4% and improve the shelter utilization efficiency. The comparision has been made in the first optimization stage. Because the number of newly-built shelters via proposed method in this study and existing shelter location model is different, obviously, the evacuation efficiency is different. To sum up, the comparison on optimization results with existing shelter location model has been completed for contruction cost in the first stage. The comparsion on evacuation efficiency is not necessary.

Point 11: It is suggested that the author show the comparison of evacuation time of 15 schemes, so as to see the difference of these results in evacuation efficiency to a certain extent.

Response 11: Table 5 is added in the revised manusript to demonstrates the comparison of evacuation time of 15 schemes.

We appreciate your professional and in-depth comments. We have made revisions based on your comments as much as possible, and will continue to improve our research framework and methods based on your comments in the future. All major revisions are highlighted in red in the revised manuscript. We sincerely thank you for your valuable comments and suggestions.

Reviewer 2 Report

The manuscript is about a combined static and dynamic optimisation method of scientifically allocating and optimizing the location of shelters needed to help with disaster management.

The manuscript is very well-written, and the research is pretty topical and timely given the increasing occurrences of natural or man-made disasters, such as the ongoing war between Russia and Ukraine. The authors have used a two-stage optimisation method taking into account the cost of construction of shelters and the evacuation time, as objective functions, and have applied circular evacuation allocation rules combined with genetic algorithms in stage 1, and used the improved social force model in stage 2.

The abstract section needs to be condensed a bit.

The introduction section presents a well-researched write-up and includes the necessary and relevant literature. The materials and methods section is too voluminous and must be condensed by trimming the detailed description of the optimisation algorithms and the process of optimisation.

The use of pictures and tables in the results and analysis section makes it easy to understand the output of the research. The discussion section summarises the novel contribution of the research and is appropriate.

The conclusion section need not summarise the study, instead highlight the academic and practical contribution of the study.

Overall, it is quality work.

Some spelling errors need to be rectified.

Please see some comments made in the attached PDF document.

Author Response

Response to Reviewer 2 Comments

Point 1: The abstract should be condensed a bit. It provides too many details upfront.

Response 1: Abstract has been condensed in the revised manuscript as follows:

Determining how to reasonably allocate shelters in the central area of the city and improve the evacuation efficiency are important issues in the field of urban disaster prevention. This paper in-troduces the methodology and mathematical model from the field of crowd emergency evacuation to shelter location optimization. Moreover, a shelter location optimization method based on the combination of static network analysis and dynamic evacuation simulation is proposed. The con-struction cost and evacuation time are taken as the objective functions. In the first stage, based on the static network analysis, a circular evacuation allocation rule based on the gravity model is proposed, and the genetic algorithm is then designed to solve the feasible schemes with the lowest shelter construction costs. In the second stage, the evacuation time is taken as the optimization objective. The age differences of refugees, the selection of evacuation routes, and the behavior of adults helping children and the elderly are simulated in a dynamic evacuation simulation model. The traditional social force model is improved to conduct a regional evacuation simulation and determine the optimal scheme with the shortest evacuation time. Finally, the central urban area of Xinyi City, Jiangsu Province, China, is taken as an empirical case.

Point 2: The introduction section presents a well-researched write-up and includes the necessary and relevant literature. The materials and methods section is too voluminous and must be condensed by trimming the detailed description of the optimisation algorithms and the process of optimisation.

Response 2: The detailed description of the optimization algorithms and optimisation process is trimmed in the revised manuscript.

Point 3: The conclusion section need not summarise the study, instead highlight the academic and practical contribution of the study.

Response 3: The academic and practical contribution of the study described on discussion in detail. Conclusion section gives the quantitative conclusions. The academic and practical contribution is highlight on conclusion section in the revised manuscript.

Point 4: (from peer-review-18024713.v2.pdf) Some spelling errors need to be rectified.

Response 4: Lines 67, “Based” is revised as “based”.

Lines 112, “Macro-models [40] analogize the movement of people to flow” is revised as “Macro-models [40] regard the movement of people as fluid”.

Lines 147, “Firstly, a regional emergency evacuation model is established via an improved social force model” is revised as “Firstly, a regional emergency evacuation model is established.”

Lines 155, “The objective function of the optimization in the first stage is considered the construction cost of shelters” is revised as “The objective function of the first optimization stage is the construction cost of shelters”.

Lines 261, “evacuation roads” is revised as “evacuation road”.

Lines 736, “combing” is revised as “combining”.

Point 5: (from peer-review-18024713.v2.pdf) Lines 238, Define WorldPop

Response 5: WorldPop(https://www.worldpop.org/) provides grid data for population density at 100m x 100m dimensions.

The above sentence is added in Lines 238.

Point 6: Lines 306, The text in this image is not legible; consider increasing the font size of the text.

Response 6: The font size of the text is increased in figure 4 in the revised manuscript.

Point 7: Lines 361, This section gives too many details about the optimisation method, which can be easily found in literature. Condense this section.

Response 7: The evacuation allocation in the first optimization stage is firstly proposed in this study, not in literature. In order to illustrate the proposed evacuation allocation model, detailed description is necessary. Some condense has been made in the revised manuscript.

Point 8: Lines 396, Condense the discussion about the research method.

Response 8: The trimming has been made in the revised manuscript from Lines 396 to Lines 415.

Point 9: Lines 482, Condense the description of the methodology. The text is too specific for general readers/audience.

Response 9: The trimming for section 4 has been made in the revised manuscript from Lines 439 to Lines 513.

We appreciate your professional and in-depth comments. We have made revisions based on your comments as much as possible, and will continue to improve our research framework and methods based on your comments in the future. All major revisions are highlighted in red in the revised manuscript. We sincerely thank you for your valuable comments and suggestions.

Reviewer 3 Report

The paper presents an interesting methodology to support evacuation planning in an urban context. The issue is relevant and the paper has numerous strengths.

However, it is necessary to improve the paper’s quality and readability. In the follow, I propose some broad and specific comments.

Author Response

Response to Reviewer 3 Comments

Point 1: Broad comments

The general considerations concern the literature review. Recent works regard action for risk reduction in transport system in emergency conditions. For instance, a recent special session of the conference “Safety and Security Engineering IX” organised by the Wessex Institute of Technology studies this problem for different point of view. I suggest the authors to introduce basic risk concepts and its components (occurrence, vulnerability and evacuation.) The paper focuses on actions to reduce exposure component.

In this case, it is important to focus on the framework for evacuation planning. A correct planning process that integrates ordinary and extraordinary conditions is useful to increase preparedness of people and managers.

I think that the paper should focus on the role of training and exercises in an integrated planning process. Different kind of exercises can improve the preparedness and the results are useful to improve the model capability to represent the phenomenon.

In this context, the current limit of the paper concerns the representation of user’s behavior. In evacuation conditions the behavior is not the same for ordinary conditions. For instance, in the proposed methodology, you adopt the gravity model approach to simulate shelter choices. Recent literature demonstrates that the specific user’s behavior in emergency conditions can be different respect to mobility in ordinary conditions (e.g. Russo and Chilà, 2014). Exercise can improve risk’s perception and capability of users to reach safe areas.

Response 1: Please provide your response for Point 1. (in red)

The basic risk concepts and its components (occurrence, vulnerability and evacuation.) is added into the revised manuscript.

On the one hand, the topic of this study is concentrated on optimization of shelter location, but not on the evacuation planning, training and exercises. This paper introduces the methodology and mathematical model from the field of crowd emergency evacuation to shelter location optimization. The methodology for optimizing the location of evacuation shelters proposed in this study combines traditional static network analysis with dynamic evacuation simulation. In the first stage, the perspective is that of government managers, so the minimum number of shelters is taken as the objective function. In the second stage, the perspective is that of the refugees. The dynamic simulation of the evacuation process determines the optimal location for the shelters to minimize the overall evacuation time.

On the other hand, this study makes a transition in the research paradigm. This study questions the assumption in traditional models of shelter location optimization that evacuees follow the routes designated by the government by taking them as the research object. The modified social force model accurately simulated the choice of evacuation path as well as the actual road congestion during the evacuation process. The change in the research paradigm proposed here has two key features. First, it replaces the traditional government-oriented planning, which follows the top-down research paradigm, with a people-oriented, bottom-up modeling method. Second, it replaces a static and linear model of evacuation to a dynamic and nonlinear model.

In fact, this paradigm shift is called for throughout schemes to optimize the distribution of shelters. For example, in the field of urban planning, in a study of the psychology of landscape preferences, Luo et al. [1] urged that the top-down, government-led planning mode be transformed into a people-oriented, top-down mode. Such a transformation is also discussed in the other urban planning research [2-4]. Therefore, in planning for the location of shelters, there is need for a comprehensive account of the demographic characteristics of the residents to be served and their evacuation behavior. A broad consensus has emerged regarding the need for a new paradigm for the optimization of shelter location, for many experts have characterized the prevailing top-down paradigm as rigid and resistant to change, which is to say, likely to continue operating in accordance with old beliefs and values despite evident problems relating to sustainability and increasingly complex societal needs. Such habitual thought patterns provide methodological stability and certainty but impede efforts to adapt the model to accommodate changes in the real world. Therefore, the new paradigm established in this study can be seen as one local facet of a broader methodological transition from a mechanistic to an organic worldview.

Thus, this study rejected the assumption that people evacuate according to the path designated by the government in the traditional location optimization of shelters, adopting instead a bottom-up research paradigm. The new research paradigm takes the refugees themselves as the research object and provides an effective crowd evacuation process based on simulation. This process then provides the basis for determining the spatial configuration of the shelters. A further departure from the traditional optimization model is the use of dynamic rather than static network analysis to simulate the actual evacuation process. Then, the approach described here replaces the traditional static and linear approach to evacuation with a dynamic and nonlinear approach. As Torres et al.[5] emphasized, this new paradigm is for a sustainable and open process, but it is not itself a goal. The new dynamic, nonlinear, bottom-up research paradigm enables the model for shelter location optimization and evacuation to adapt to complex real-world situations.

Reference

[1] Luo T, Xu M, Liu J, Zhang JQ (2019) Measuring and Understanding Public Perception of Preference for Ordinary Landscape in the Chinese Context: Case Study from Wuhan. Journal of Urban Planning and Development 145(1): 05018021, DOI: 10.1061/(ASCE)UP.1943-5444.0000492

[2] Martins MS, Fundo P, Kalil RML, Rosa FD (2021) Community participation in the identification of neighbourhood sustainability indicators in Brazil. Habitat International 113(1):102370, DOI: 10.1016/j.habitatint.2021.102370

[3] Mustaffa NK, Isa CM, Ibrahim CK (2021) Top-down bottom-up strategic green building development framework: Case studies in Malaysia. Building and Environment 203(7):108052, DOI: 10.1016/j.buildenv.2021.108052

[4] Guo J, Zeng Y, Zhu K, Tan X (2021). Vehicle mix evaluation in Beijing’s passenger-car sector: From air pollution control perspective. Science of the Total Environment 785:147264, DOI: 10.1016/j.scitotenv.2021.147264

[5] Franco-Torres M, Rogers BC, Harder R (2021). Articulating the new urban water paradigm. Critical Reviews in Environmental Science and Technology 51: 2777-2823, DOI: 10.1080/10643389.2020.1803686

Point 2: Figure 4 is not fully readable; some characters are not readable. Please try to simplify the

figure to generalize the approach. If it is necessary, you can specify each step with single figures.

Response 2: Figure 4 is updated in the revised manuscript. The reader/audience can understand the process of methodology combining figure 4 with the following description.

(1) Basic data acquisition. The road hierarchy, the spatial location of evacuation demands, and the shelters in the study area are determined from remote sensing images, Google Maps, OpenStreetMap [65], and field research. Some scholars have determined the spatiotemporal distribution of the population based on big data technologies such as point-of-interest (POI) data [66], mobile signaling data [67], Baidu heat maps [68], and social media sign-in data [69]. In the present study, the 24-h population change curve and dynamic evacuation demands in different periods are determined based on mobile phone signaling data. Moreover, the age structures of refugees are determined by data from WorldPop (https://www.worldpop.org/).

(2) The optimization in the first stage aims at minimizing the number of shelters via static network analysis. First, the circular evacuation allocation rule is proposed. Circular evacuation allocation indicates that refugees at the demand point are allocated to the surrounding shelters based on the gravity model. If the number of refugees received by one shelter reaches the capacity limitation of the shelter, the evacuation will stop at this time. Pedestrians who have not been evacuated will enter the next cycle until all people have completed the evacuation. Second, under the constraints of the capacity and service distance, a genetic algorithm is designed to solve the minimum number of newly-built shelters. Third, the exhaustive method is adopted to select the feasible schemes that satisfy the dynamic evacuation demands of all 24 hours and the limitations of the capacity and service distances for  (B is the total number of candidate shelters) types of candidate schemes.

(3) Establishment and improvement of the regional emergency evacuation model. Considering the choice of evacuation routes, this study improves the traditional social force model and establishes a selection function for evacuation routes. Furthermore, the spatial heterogeneity of the age structure of pedestrians and the behavior of adults helping children and the elderly during the evacuation process are considered. The simulation platform Anylogic is used to simulate the evacuation process to determine the total evacuation time and identify the congested road sections in the study area.

(4) In the second stage, the evacuation time is taken as the objective function for optimization, and the emergency evacuation simulation of the feasible schemes identified in the first stage is completed. The optimal shelter location scheme with the minimum number of shelters and the shortest evacuation time is then determined.

Point 3: Some basic assumptions are not fully clear; please clarify the assumptions: Do they derive

from literature review?

Response 3: Assumption (2) is clarified in the revised manuscript. Assumptions derive from reference [70][71][72].

[70] Ministry of Housing and Urban-Rural Construction of the People's Republic of China. GB/T 51328-2018 Standard for urban comprehensive transport system planning. Beijing: China Architecture & Building Press, 2018. (in Chinese).

[71] Ministry of Housing and Urban-Rural Construction of the People's Republic of China. CJJ 37-2012 Code for design of urban road engineering. Beijing: China Architecture & Building Press, 2016. (in Chinese).

[72] Xu, X.; Tang. Q. Urban Road and Transportation Planning; China Architecture & Building Press: Beijing, China, 2007; ISBN 7112075955.

Point 4: Please, avoid reporting numerical results in the abstract

Response 4: Abstract is trimmed as follows in the revised manuscript. The numerical results is deleted.

“Determining how to reasonably allocate shelters in the central area of the city and improve the evacuation efficiency are important issues in the field of urban disaster prevention. This paper in-troduces the methodology and mathematical model from the field of crowd emergency evacuation to shelter location optimization. Moreover, a shelter location optimization method based on the combination of static network analysis and dynamic evacuation simulation is proposed. The con-struction cost and evacuation time are taken as the objective functions. In the first stage, based on the static network analysis, a circular evacuation allocation rule based on the gravity model is proposed, and the genetic algorithm is then designed to solve the feasible schemes with the lowest shelter construction costs. In the second stage, the evacuation time is taken as the optimization objective. The age differences of refugees, the selection of evacuation routes, and the behavior of adults helping children and the elderly are simulated in a dynamic evacuation simulation model. The traditional social force model is improved to conduct a regional evacuation simulation and determine the optimal scheme with the shortest evacuation time. Finally, the central urban area of Xinyi City, Jiangsu Province, China, is taken as an empirical case.”

Point 5: There are some physical limits that could play a relevant role in evacuation time? For instance the physical dimensions of urban road network (e.g. barriers for users with special needs); in this case the route choice is influenced by these limits

Response 5: In the evacaution model of this study, “Research on crowd evacuation behavior shows that when a disaster occurs, the choice for evacuation routes is influenced by the distances from demand points to shelters, the population density, and the width of evacuation roads [78]”.

The physical dimensions of urban road network is considered in the assumption (2).

Reference [78] in the revised manuscript:

Yu, J.; Zhang, C.; Wen, J.; Li, W.; Liu, R.; Xu, H. Integrating multi-agent evacuation simulation and multi-criteria evaluation for spatial allocation of urban emergency shelters. International Journal of Geographical Information Science 2018, 32(9), 1884-1910.

Point 6: It is not clear if you consider or not the role of congestion in the calculation of evacuation

time

Response 6: The congestion of evacuation roads is considered, which has been emphasized on Lines 261, Lines 264, Lines 301, Lines 319 and Lines 687 in the revised manuscript.

Point 7: Some formulations are not fully readable (e.g. row 267)

Response 7: The explaination is added in the revised manuscript.

The number of candidate shelters is m, i is the serial number of candidate shelter, so the total number of shelters location schemes is determined as .

Point 8: Suggested references

Response 8: Yes. The following references are added in the revised manuscript.

[2] Russo, F.; Rindone, C. Urban exposure: Training activities and risk reduction. WIT Transactions on Ecology and the Environment 2014, 191, 991-1001.

[52] Russo, F.; Chilà, G. A prototypal test using stated preferences data to model evacuation decisions. WIT Transactions on Ecology and the Environment 2013, 173, 743-752.

[80] Russo, F.; Chilà, G. Integrated travel demand models for evacuations: a bridge between social science and engineering. Inter-national journal of safety and security engineering 2014, 4(1), 19-37.

We appreciate your professional and in-depth comments. We have made revisions based on your comments as much as possible, and will continue to improve our research framework and methods based on your comments in the future. All major revisions are presented by “Track Changes” in the revised manuscript. We sincerely thank you for your valuable comments and suggestions.

Reviewer 4 Report

The work is detailed and informative. I appreciate it a lot. However, there is still room to improve it. I mentioned the points below.

In the first paragraph of the Introduction, lines 31-41, “With the development of natural and social systems becoming increasingly more complex, the frequency and types of disasters are increasing.”, the sentence is the same as the first sentence of the abstract. Please rephrase it and not repeat it.

Lines 95-97, “However, due to the separation of workplaces and residences in central urban areas, there are substantial differences in the population distributions during daytime and nighttime.” Is there any evidence to demonstrate that the population in central urban areas are different during daytime and nighttime?

From lines 165-171, this paragraph is about the content of this manuscript. This information can be deleted since it is not quite important, and the readers would like to know more about the contributions of this study. Therefore, the contribution of this study can be placed there instead of the contents of sections 2, 3, 4…

For Lines 180-182, the reference related to the 8.5-magnitude earthquake in 1668 had to be cited so that reader can refer to the information.

Lines 185-186, “the population distribution varies greatly between daytime and nighttime.” Evidence should be provided to prove it.

What are the relevant references for equations? Some are missing. Please check and add back.

I believed this study is not without limitations. Please add this section at the end of the Discussion section.

Minor issue:

Lines 227-228, there is only one sentence in one paragraph. Please improve these kinds of structures.

Line 67, “For instance, Based on…”, B in “Based” should be lowercase, “based”. Please check the whole article and revise all these types of mistakes.

Author Response

Response to Reviewer 4 Comments

Point 1: In the first paragraph of the Introduction, lines 31-41, “With the development of natural and social systems becoming increasingly more complex, the frequency and types of disasters are increasing.”, the sentence is the same as the first sentence of the abstract. Please rephrase it and not repeat it.

Response 1: This sentence “With the development of natural and social systems becoming increasingly more complex, the frequency and types of disasters are increasing” has been deleted from abstract in the revised manuscript, so this sentence in introduction can be reserved.

Point 2: Lines 95-97, “However, due to the separation of workplaces and residences in central urban areas, there are substantial differences in the population distributions during daytime and nighttime.” Is there any evidence to demonstrate that the population in central urban areas are different during daytime and nighttime?

Response 2: The evidence is illustrated in the following references:

Qi, W.; Li, Y.; Liu, S.; Gao, X.; Zhao, M. Estimation of urban population at daytime and nighttime and analyses of their spa-tial pattern: A case study of Haidian District, Beijing. ACTA GEOGRAPHICA SINICA 2013, 68(10), 1344-1356.

Akkerman, A. The urban household pattern of daytime population change. The Annals of Regional Science, 1995, 29(1): 1-16.

Zhong, W.; Wang, D.; Xie, D.; Yan, L. Dynamic characteristics of Shanghai's population distribution using cell phone signaling data. GEOGRAPHICAL RESEARCH 2017, 36(5), 972-984.

These references is added in the revised manuscript.

Point 3: From lines 165-171, this paragraph is about the content of this manuscript. This information can be deleted since it is not quite important, and the readers would like to know more about the contributions of this study. Therefore, the contribution of this study can be placed there instead of the contents of sections 2, 3, 4…

Response 3: Yes. The contents of sections 2, 3, 4… is replaced by contributions in the revised manuscript:

The proposed methodology of this study makes four contributions: (1) introduces the methodology and mathematical model from the field of crowd emergency evacua-tion to shelter location optimization. The deficiencies in the configuration of urban shelters can be intuitively found; (2) the combined static/dynamic optimization meth-odology cannot only efficiently solve the location optimization, but also reflects the authentic crowd evacuation process; (3) the dynamic evacuation demands are accu-rately determined which can satisfy the evacuation demand of residents at all times; (4) the improved evacuation model simulates the behavior of adults helping the elderly and children, choice behavior for evacuation routes, and spatial heterogeneity of the age structure.

Point 4: For Lines 180-182, the reference related to the 8.5-magnitude earthquake in 1668 had to be cited so that reader can refer to the information.

Response 4: Yes. The following reference is added in the revised manuscript.

Gu, Q.; Xu, H.; Yan, Y.; Zhao, Q.; Li, L.; Meng, K.; Yang, H.; Wang, J.; Jiang, X.; Ma, D. The crustal shallow structures and fault activity detection in Xinyi section of Tanlu Fault Zone. Seismology and geology 2020 42(4), 825-843.

Point 5: Lines 185-186, “the population distribution varies greatly between daytime and nighttime.” Evidence should be provided to prove it.

Response 5: In response 2, relevant references added in the revised manuscript.

Point 6: What are the relevant references for equations? Some are missing. Please check and add back.

Response 6: The following refercence is added in equation (1) in the revised manuscript.

[1] Ministry of Housing and Urban-Rural Construction of the People's Republic of China. GB51143-2015 Code for design of disas-ters mitigation emergency congregate shelter. Beijing: China Architecture & Building Press, 2015. (in Chinese).

The following refercence is added in equation (19) in the revised manuscript.

[73] Yang, J.; Hu, Y.; Zhang, K.; Wu, Y. An improved evolution algorithm using population competition genetic algorithm and self-correction BP neural network based on fitness landscape. Soft Computing 2021, 25, 1751-1776.

The following refercence is added in equation (20) and (21) in the revised manuscript.

[74] Helbing, D.; Farkas, I.; Vicsek, T. Simulating dynamical features of escape panic. Nature 2000, 407, 487-490.

The other equations are established in this study, which are not from literatures.

Point 7: I believed this study is not without limitations. Please add this section at the end of the Discussion section.

Response 7: The limitation of this study is added in the revised manuscript.as follows:

Although this study has made four contributions, the proposed model still has some deficiencies:

(1) People’s preferences for different types of shelters is not considered in evacua-tion model.

(2) All evacuation roads are assumed to be unobstructed, which is inconsistent with actual situation. After earthquake occurs, some evacuation routes will be blocked by debris of buildings.

Determining how to simulate people’s preferences for different shelters and how to consider the influence of the disaster environment on the evacuation process will be the focus of future research. We can bring the risk of road blocking and building dam-age caused by hazards into the evacuation scenario by combining the advanced disas-ter simulation technology with regional evacuation simulation, so as to determine the evacuation demands more accurately and reflect the evacuation process more authen-tic. The scientism and accuracy of the evacuation process and the shelter location op-timization will be improved.

Point 8: Lines 227-228, there is only one sentence in one paragraph. Please improve these kinds of structures.

Response 8: The sentence “Table 2 demonstrates the number of refugees at each moment and the age structure of the demand points” is put on the end of the previous paragraph in the revised manuscript.

Point 9: Line 67, “For instance, Based on…”, B in “Based” should be lowercase, “based”. Please check the whole article and revise all these types of mistakes.

Response 9: Line 67, Based on…” is correted as “based” in the revised manuscript.

We appreciate your professional and in-depth comments. We have made revisions based on your comments as much as possible, and will continue to improve our research framework and methods based on your comments in the future. All major revisions are highlighted in red in the revised manuscript. We sincerely thank you for your valuable comments and suggestions.

Round 2

Reviewer 1 Report

Line 661,Why is the minimum value in Table 5 not 1802 seconds corresponding to line 661?

Formula (4) is unnecessary, and there is no place to use Ri

Reviewer 3 Report

The paper can benaccepted

Reviewer 4 Report

n/a